# Selective flexible packaging pathways of the segmented genome of influenza A virus

Ivan Haralampiev[1,8,11], Simon Prisner[1,11], Mor Nitzan [2], Matthias Schade[1], Fabian Jolmes [1,9], Max Schreiber [3], Maria Loidolt-Krüger[4,9], Kalle Jongen[1], Jasmine Chamiolo[5], Niklaas Nilson[1], Franziska Winter [4,10], Nir Friedman [2,6], Oliver Seitz [5], Thorsten Wolff [7✉] & Andreas Herrmann [1✉]

The genome of influenza A viruses (IAV) is encoded in eight distinct viral ribonucleoproteins (vRNPs) that consist of negative sense viral RNA (vRNA) covered by the IAV nucleoprotein. Previous studies strongly support a selective packaging model by which vRNP segments are bundling to an octameric complex, which is integrated into budding virions. However, the pathway(s) generating a complete genome bundle is not known. We here use a multiplexed FISH assay to monitor all eight vRNAs in parallel in human lung epithelial cells. Analysis of $3.9 \times 10^5$ spots of colocalizing vRNAs provides quantitative insights into segment composition of vRNP complexes and, thus, implications for bundling routes. The complexes rarely contain multiple copies of a specific segment. The data suggest a selective packaging mechanism with limited flexibility by which vRNPs assemble into a complete IAV genome. We surmise that this flexibility forms an essential basis for the development of reassortant viruses with pandemic potential.

[1] Molecular Biophysics, Department for Biology, IRI Life Sciences, Humboldt-Universität zu Berlin, Invalidenstr. 42, 10115 Berlin, Germany. [2] School of Computer Science and Engineering, The Hebrew University of Jerusalem, Jerusalem 91904, Israel. [3] Competence Centre Biomedical Data Science, Institute for Applied Informatics e.V. at Leipzig University, Goerdelerring 9, 04109 Leipzig, Germany. [4] Department of NanoBiophotonics, Max-Planck-Institute for Biophysical Chemistry, Am Fassberg 11, 37077 Göttingen, Germany. [5] Bioorganic Synthesis, Department for Chemistry, Humboldt-Universität zu Berlin, Brook-Taylor-Straße 2, 12489 Berlin, Germany. [6] Institute of Life Sciences, The Hebrew University of Jerusalem, Jerusalem 91904, Israel. [7] Robert Koch Institut, Unit 17, Influenza and Other Respiratory Viruses, Seestraße 10, 13353 Berlin, Germany. [8] Present address: Crystallography, Max Delbrück Center for Molecular Medicine, Robert-Rössle Str. 10, 13125 Berlin, Germany. [9] Present address: PicoQuant, Rudower Chaussee 29, 12489 Berlin, Germany. [10] Present address: Abberior Instruments GmbH, Hans-Adolf-Krebs-Weg 1, 37077 Göttingen, Germany. [11] These authors contributed equally: Ivan Haralampiev, Simon Prisner. ✉email: WolffT@RKI.de; andreas.herrmann@rz.hu-berlin.de

nfluenza A viruses (IAV) belong to the *Orthomyxoviridae* family and are the causative agents of human influenza, a respiratory disease with up to 646,000 deaths annually by seasonal infections[1]. The IAV genome is encoded in eight negatively orientated viral RNAs (vRNA) ranging from 0.9 to 2.3 kb in length. They are organised as viral ribonucleoproteins (vRNP), which are decorated with viral nucleoproteins (NP) and a single viral RNA-dependent RNA polymerase (RdRp)[2]. New vRNPs are formed in the nucleus and transported via the CRM1 nuclear export machinery to the cytosol. Rab11-positive endomembrane organelles[3,4] carry them towards the plasma membrane, where they are incorporated as vRNP bundles into assembling virions. The segmented nature of the IAV genome can, on one hand, provide an evolutionary benefit as it enables the virus to evolve by reassortment of gene segments. On the contrary, reassortment may also bring together viral segments encoding proteins from parental strains, which work less well together, thereby reducing viral fitness[5,6] (for a review see ref. [7]). Hence, knowledge of the mechanism(s) by which vRNPs assemble into a genome complex may support an improved understanding of biological phenotypes conferred by viral reassortment.

Random and selective packaging models have been proposed to explain assembly of eight distinct vRNPs into virions[8]. Note that strictly speaking, 'packaging' refers to the incorporation of the vRNP segment bundle into the emerging virion, while 'bundling' describes the process of the vRNP bundle assembly. However, as packaging is used very often synonymously for only the bundling in literature, we will do so here as well. The random model presumes that vRNP segments interact in a stochastic manner, leading to bundles and virions that do not necessarily contain a complete set of vRNPs[9], and may contain multiple copies of one segment. However, recent reports employing different techniques[10–15] provide solid evidence for the selective packaging model: the different vRNPs interact specifically with each other and thus ensure the incorporation of exactly one copy of each segment into multi-segment complexes (MSC). Using length as a parameter to distinguish between vRNP species, electron tomography of intact virions[10,12] suggests that the viral genome is organised as an MSC with eight different segments, 7 of which are arranged around a central segment in a '7 + 1' pattern[2,16]. Evidence for selective packaging includes the identification of packaging signals in conserved 3′ and 5′ terminal non-coding (NCRs) and coding regions (CRs) of the vRNAs, mediating interactions between vRNPs[13,15,17–27]. Those CRs serve as segment-bundling signals, whereas the NCRs operate as incorporation signals. Bundling sequences are proposed to interact with each other, whereby similar signals compete for integration and thus, ensure formation of bundles consisting of exactly eight different segments. Incorporation signals within the NCRs assure integration of the corresponding segment into progeny viruses. Fluorescence in situ hybridisation (FISH) studies[28,29] resolving in parallel up to four out of the eight distinct vRNAs provide evidence that these signals lead to formation of MSCs along the transit of the segments to the plasma membrane of infected cells.

Although there is broad consensus on the non-random, selective packaging model of viral genome formation, the assembly pathway(s), i.e., the order in which segments form a complete octameric MSC, has not yet been elucidated[8]. An important step towards decoding the assembly pathway would be the availability of an approach to visualise and distinguish the eight vRNPs in infected single cells.

Here, we apply a Multiple Sequential FISH-assay (MuSeq-FISH) to simultaneously visualise and distinguish all eight vRNA segments and their presence in MSCs in single permissive, IAV-infected human respiratory cells in a quantitative manner. Using spinning-disc microscopy to detect vRNA spots and their colocalisation, we are able to extract the segment composition of about 10^5 MSCs. Analysis of the segment composition of intermediate MSCs in infected cells is indicative of a flexible selective packaging mechanism by which vRNPs are assembled into a complete octameric IAV genome complex. Indeed, the major fraction of mature virions contains one copy of each of the eight vRNPs. These results, together with the observed impairment of vRNP bundling upon non-permissive IAV infection, provide a framework for future work to decipher the precise rules of segment bundling and gene reassortment, and the involvement of host-cell factors.

## Results

**Imaging of IAV vRNAs and vmRNAs.** We studied vRNP bundling in human A549 cells infected with the prototypic seasonal influenza A/Panama/2007/99 virus (H3N2) at a single-cell level after 10 h.p.i. (MOI 5). To visualise all vRNAs and major viral mRNAs (vmRNAs), we applied a multiple sequential FISH assay[30] (MuSeq-FISH, see 'Methods') very similar to a recently published procedure[31]. Up to 12 MuSeq-FISH cycles were performed, each simultaneously targeting two RNA species with probe sets coupled to two different fluorophores (Atto550, Fig. 1a; STAR635P, Supplementary Fig. 1, Supplementary Table 1). After each of the cycles, probes were removed by formamide treatment (see 'Methods'; Supplementary Figs. 2–4). To reduce potential false-positive signals, each vRNA was targeted twice along the cycles, once with Atto550 (Fig. 1a) and once with STAR635P-labelled probe sets (Supplementary Fig. 1). Cycles for the first and second labelling of a given vRNA were randomly selected among the 12 cycles, and varied between triplicates to exclude a potential bias introduced by the order of staining. We excluded vRNA spots observed only for one probe set from further analysis.

Microscopic analysis showed that the in situ staining patterns of vRNAs and of NP, as detected by an antibody, were almost identical, demonstrating the specificity of FISH probes for vRNP segments (Fig. 1b, α-NP + vRNA). High degree of colocalisation was observed for all vRNA segments (Fig. 1a, overlay, white colouring). We tested probe specificity of A/Panama probes targeting A549 cells infected with the influenza B/Lee virus (see Supplementary Fig. 5). Any fluorescence signal arising even from unspecific binding of FISH probes could not be detected. Negative results for these probes were also obtained for mock-infected A549 cells. To further verify probe specificity, we compared the intracellular distribution of vRNAs with that of all major viral mRNAs (vmRNAs) (Fig. 1c). Cytosolic vmRNA spots did not significantly colocalise with vRNA (Fig. 1 and Supplementary Fig. 1) and with each other (see below). For more details on imaging of vmRNAs and viral cRNAs, see Supplementary Note 1.

We analysed a total of 69 A/Panama-infected cells, yielding more than $3.9 \times 10^5$ vRNA spots, each of which corresponded to the detection of a single RNA segment (Fig. 2a). Nuclear vRNA spots were excluded for analysis for several reasons: (i) low amounts of segments found in the nucleus (Fig. 1), (ii) incomplete vRNAs produced inside the nucleus and detected by FISH may not be capable of interacting with other segments and (iii) bundling is presumed to take place in the cytosol[29]. Consistent with a previous report[32], we observed cell-to-cell variability in the absolute number of individual segments and in the normalised number of individual segments as revealed by the frequency distribution of specific segments (Supplementary Fig. 7) and statistical analysis (Fig. 2, legend, Bartlett test), respectively. Nevertheless, for the majority of cells, at the individual cell level, a balanced ratio between the eight vRNA segments was observed (Fig. 2a) with no significant difference between the segments (ANOVA, F test, for details see legend to Fig. 2).

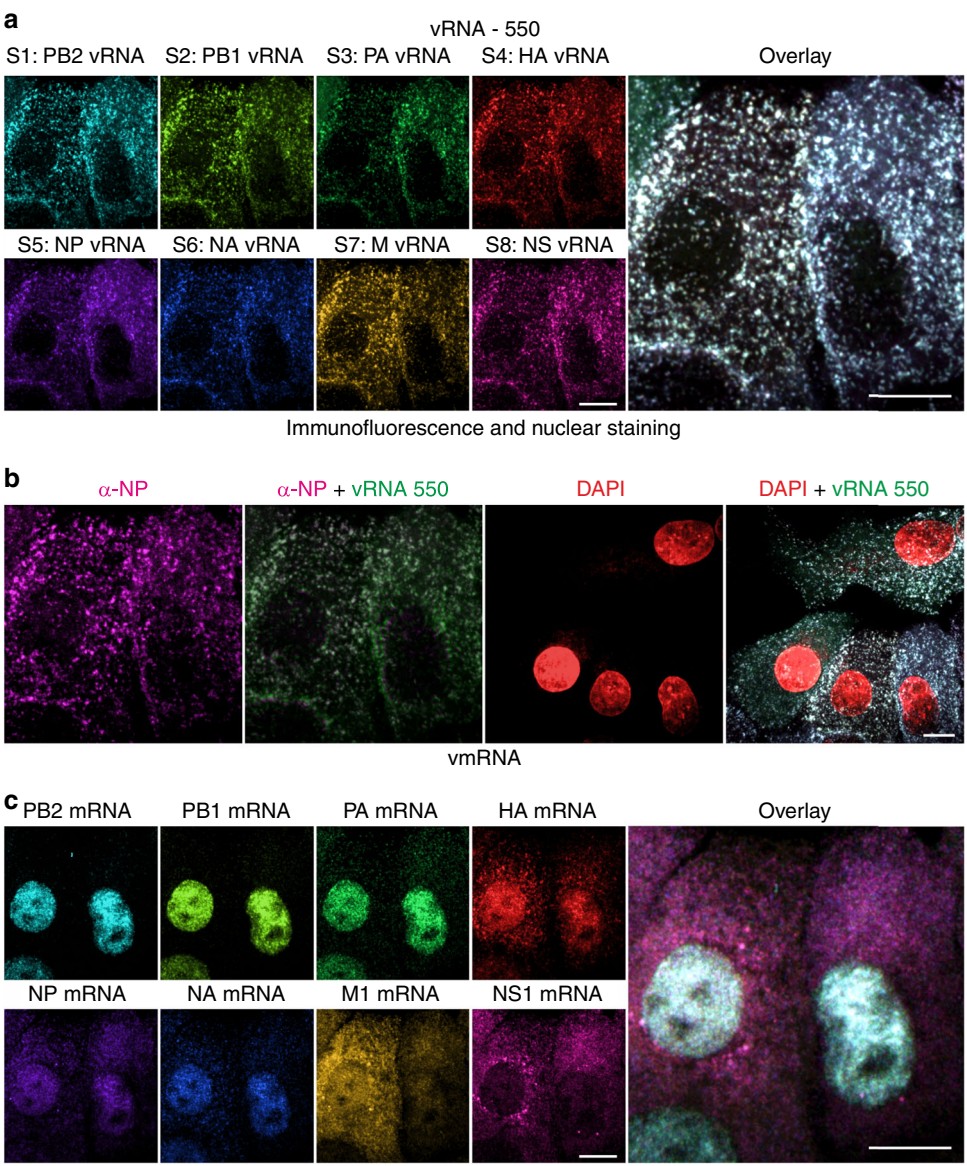

**Fig. 1 Localisation of viral RNA (vRNA) and vmRNA in A/Panama-infected A549 cells. a** Viral genomic RNAs were stained by fluorescence in situ hybridisation (FISH) 10 h.p.i. (MOI 5). To cover all vRNAs and major vmRNAs, 12 cycles of FISH labelling were performed. Along these cycles, each vRNA was targeted twice, once with Atto550 (shown here) and once with STAR635- (Supplementary Fig. 1b) labelled probe sets (for details see text). **b** Immunofluorescence staining of NP. DAPI (4′,6-Diamidin-2-phenylindol) labelling was performed to exclude nuclear vRNA spots for further colocalisation analysis. **c** All major unspliced vmRNAs were stained. Images represent max z projections. The intensities of images were scaled according to the corresponding images taken after probe removal by formamide. Scale bars correspond to 10 μm. Representative images of four independent experiments are shown.

Overall, the population-averaged fractions of the various vRNAs were equal, with each appearing at about 12.5% (Fig. 2b; statistics, see legend). This agrees with qRT-PCR measurements at the cell population level (Supplementary Fig. 8) showing that the expression levels of the different vRNA species were similar. We also studied vRNA formation in A549 cells infected with A/Mallard/439/2004, a low-pathogenic avian IAV that is dicussed below.

**Analysis of rank and segment composition of MSCs.** To relate the vRNA signals to each other, we initially conducted a colocalisation analysis binning two or more vRNA spots within a cylinder of radius 300 nm and height 1000 nm into one MSC (Supplementary Note 2). Although the size of isolated vRNP complexes varying in length between 50 and 150 nm[33] is smaller,

these cylinder dimensions were chosen for two reasons: the resolution limit of confocal fluorescence microscopy being roughly 200 nm[34] in the $x$–$y$ direction, and findings by Chou et al. and Lakdawala et al.[28,29] who have used a radius of 255 nm in the $x$–$y$ direction comparable to our cylinder radius. Our 3D-stack acquisition in the z direction was performed with 400-nm steps due to the lower resolution in the z direction of confocal microscopy, which is the reason for choosing a height of 1000 nm for colocalisation analysis. Significantly, a careful assessment of the informative value of this analysis involving high-resolution stimulated emission depletion (STED) microscopy provided strong support for the validity of this approach (see below). All nuclear spots were excluded on the premise that segment bundling takes place in the cytosol[29], as confirmed by the abundant presence of MSC of lower ranks, and predominant colocalisation

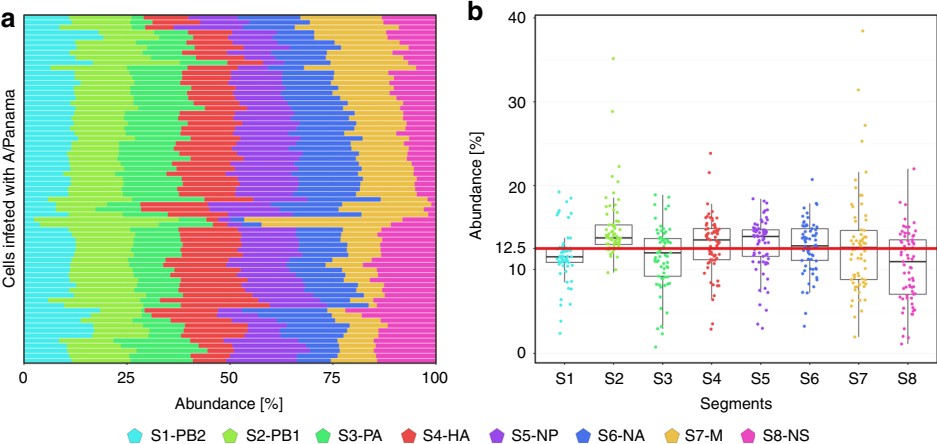

**Fig. 2 Viral RNA (vRNA) abundances in A/Panama-infected A549 cells 10 h.p.i. a** Relative abundances of all vRNA segments per cell as measured by fluorescence in situ hybridisation (FISH). For each of the measured cells ($n = 69$ from four independent experiments), the number of spots that were detected was counted and normalised to the total number of spots per cell. The colouring scheme is the same as in Fig. 1. **b** Data shown in **a** in a Tukey boxplot representation. Upper bound of box, centre and lower bound of the box represent the 75th percentile, the 50th percentile (median) and the 25th percentile, respectively. Upper and lower whiskers represent the maxima and minima of the boxplots showing the respective largest or smallest value within 1.5× interquartile range above the 75th or below the 25th percentile (bounds of box). In all, 12.5% (red line) corresponds to the expected fraction for each vRNA in a uniform distribution. For the segments, there is no significant difference (ANOVA, $F$ test, one-sided, $\alpha = 0.05$) between the cells ($P > 0.999$). The application of ANOVA requires Gaussian distribution (fulfilled, see Supplementary Fig. 7) and the same variability (standard deviation) for the segments. However, the Bartlett test (one-sided, $\alpha = 0.05$) shows that the variability of the individual segments is significantly different ($P > 0.0001$). The Kruskal–Wallis test does not presuppose homogeneity of variability, but agrees with the $F$ test in the result, i.e., no significant differences (one-sided, $P > 0.999$). Source data are provided as a Source Data file.

of vRNA spots in this compartment (Fig. 1). In all, i.e., 69 cells, a total of about $1 \times 10^5$ MSCs were found, 69% of which represented multimeric complexes.

We found only in less than 5% indications for the presence of two, but not more copies of the same species. However, in view of the optical resolution achieved, our approach does not allow to address accurately whether multiple copies of a segment are present in an MSC. Furthermore, Chou et al.[35] have shown that not all specific FISH probes bind to a segment of purified viruses. Thus, we cannot assume that all of the FISH probes we have used for a specific segment will bind. In particular, the accessibility of a segment could be additionally impacted in infected cells, and the presence of more than one copy of the same segment might be missed. Nevertheless, an earlier report on competition between vRNAs that possess the same packaging signal, but encode different reporter genes, suggested inefficient incorporation of multiple copies of the same vRNP in one MSC[36]. In fact, considering all possible combinations of vRNPs (see Supplementary Methods and Supplementary Eqs. 1–3), the random packaging model predicts that 98% of the MSCs would contain two or more copies of a distinct segment species. Hence, the observed low prevalence of two segment copies within one MSC is in line with a selective vRNP packaging model with a controlled segment composition of MSCs. Based on these findings and previous reports[10–13,28,29,35], we considered that only one copy of each vRNA segment is packaged into an MSC and ranked 8 as the highest rank of MSCs corresponding to an IAV genome with a complete vRNP set.

In Fig. 3, the frequency distribuiton of MSC ranks (a–d), the total number of segments in infected cells (e–h) and the number of segments not involved in MSC formation (i–l) are shown. Typically, we observed a marked u-shaped frequency distribution of MSC ranks with the two maxima representing free segments (monomers, rank 1) and octameric complexes (rank 8), respectively (Fig. 3a, b, filled bars). 'Cell 18' illustrates this in an exemplary way (see 'Rank Distribution', Fig. 3b, filled bars). Significantly, these features and particularly the u-shaped distribution of MSC ranks described above were also observed

when averaging over all cells (Fig. 3a), with only a few cells deviating from this pattern (see below). These rank distributions are significantly different from a uniform distribution (Chi-square test, $\alpha = 0.05$; $P < 0.0001$). The amounts of S1–S8 segments (Fig. 3e, 'Total Segments') were almost equal, except for small, but significant differences between S2 and S8 and S5 and S8 (statistics, see legend to Fig. 3) (see 'Discussion').

MSCs of high rank made up the majority of the total segments found (Fig. 3a, b, empty bars), and the number of solitary segments (rank 1) was low (Fig. 3i, j) showing that the majority of each segment assembles into MSCs. This observation strongly suggests that interactions between the segments direct towards the formation of high-rank MSCs. For a few cells (Cell 23 (Fig. 3c) and Cell 29 (Fig. 3d)), rank distributions were significantly different from those in Fig. 3a and b (Chi-square test, $\alpha = 0.05$; $P < 0.0001$), which may provide clues for MSC assembly. Notably, the amounts of segment 8 (Cell 23 (Fig. 3g) and Cell 29 (Fig. 3h)) and segment 1 (Cell 23 (Fig. 3g)) were low in comparison to the remaining segments (see 'Discussion').

Previous studies[28,29] have reported an increase in the frequency of complexes with increasing rank in infected cells. At a first glance, this seems to be in contrast to the u-shaped distribution observed here. However, this apparent discrepancy to the earlier studies can easily be reconciled by the inherent limitation to resolve only a maximum of four distinct segments. Consequently, since no complexes of rank 5 or higher could be identified, MSCs observed as rank 1–4 also include complexes ranked higher than rank 4, leading the authors to suggest a continuous increase in the frequency of complexes with increasing rank.

Next, we focused on the segment composition of MSCs at ranks 2–7 (Fig. 4), which most likely represent intermediate complexes of the assembly pathway (see below). A striking finding was that for any given intermediate rank, there was not a single specific segment composition of high abundance, but several, more quantitative than qualitatively preferred, partly overlapping segment combinations. This observation indicates a limited variability of specific interactions between the vRNPs, but

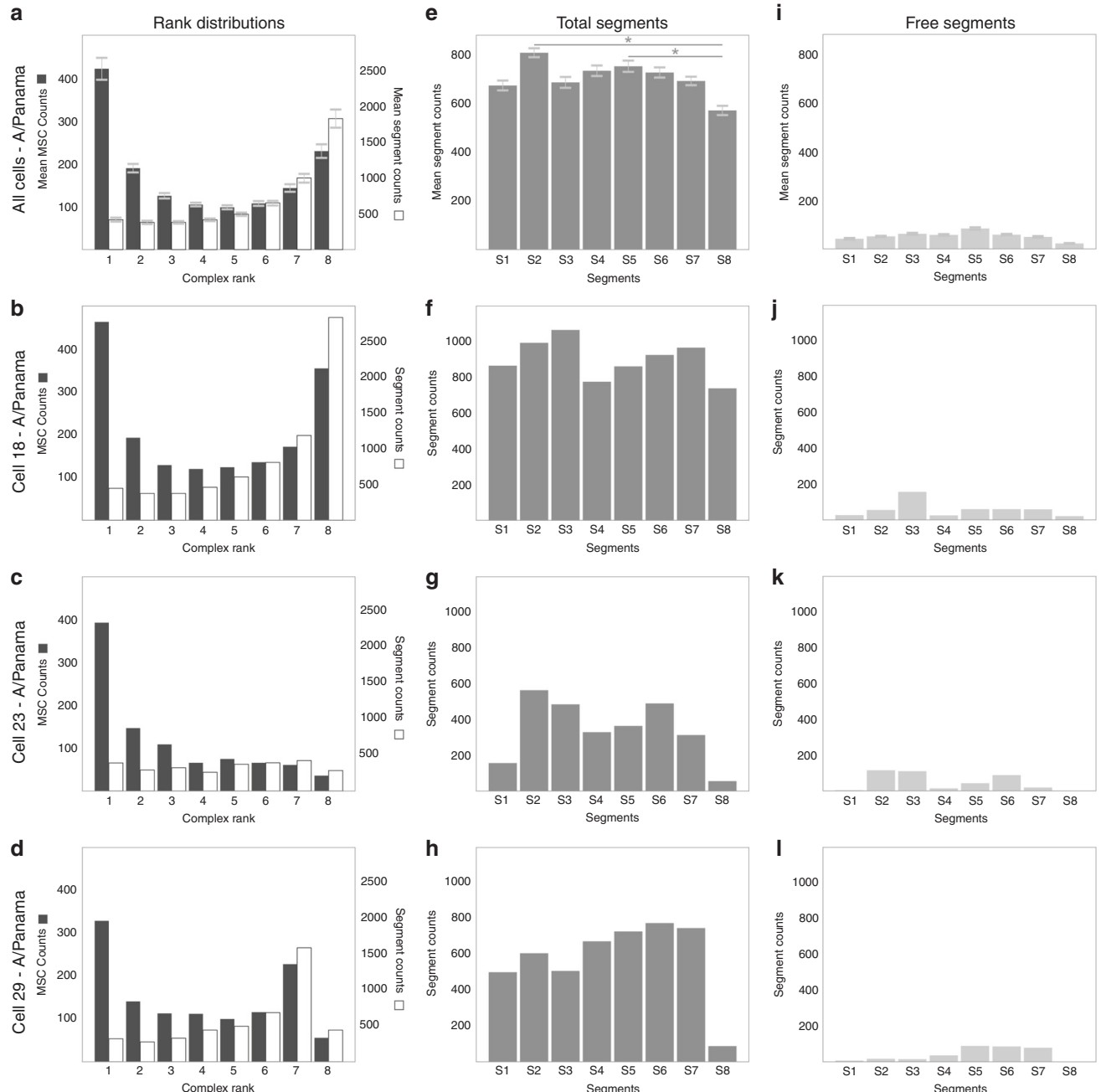

**Fig. 3 Distribution of MSC ranks in A/Panama-infected A549 cells after 10 h.p.i.** A total of $3.9 \times 10^5$ distinct segment spots were detected in A/Panama-infected A549 cells and binned into about $1 \times 10^5$ MSCs. The frequency distributions of MSC ranks (**a–d**, 'Rank Distributions', filled bars: 'MSC counts', empty bars: 'Segment counts'), of all segments S1–S8 occurring in a cell (**e–h**, 'Total Segments') and of free, i.e., monomeric segments, i.e., rank 1 (**i–l**, 'Free Segments') are shown. Frequency distributions for all 69 cells (**a**, **e**, **i**, 'All Cells') and exemplary for a single cell (e.g., 'Cell 18' **b**, **f**, **j**). Frequency distributions for cells in which particular segments appeared in unusually low quantities: counts were low for S1 and S8 in 'Cell 23' **c**, **g**, **k**, and for S8 in 'Cell 29' (**d–l**). In **a**, values are shown as mean ± s.e.m. Statistics for 'Total Segments': **a** multiple comparison of means by Tukey test (one-sided, $\alpha = 0.05$) reveals the following significant differences between segments: **a** S2–S8 and S5–S8 ($P = 0.041$; *$P \leq 0.01$); Source data are provided as a Source Data file.

argues against a strict, non-variable sequence of vRNP interactions consistent with recent experimental and modelling reports[14,37,38]. In the latter case, we would have expected a single distinct MSC composition to stand out per MSC rank.

In order to get a first impression of the dynamics of MSC formation, we analysed compositions of MSCs after 6 h.p.i. (MOI 5). The frequency distribution of the observed ranks of MSCs in the cytosol 6 h.p.i. (Supplementary Fig. 9) showed a left-peaked form. It is obvious that the higher ranks at this time after infection are only present in small amounts. A comparison with

the results obtained 10 h.p.i. (Fig. 3a) shows the dynamic character of the formation of the MSCs. Although further investigations are required, one reason for the reduced frequency of high-rank MSCs may be due to the lower number of segments per cell ($3.1 \times 10^3$ (6 h.p.i.) vs. $5.6 \times 10^3$ (10 h.p.i.)).

**vRNP segments in intact IAV virions**. To assess the vRNP composition of mature A/Panama virions by MuSeq-FISH, virions present in virus stocks were allowed to bind to the surface of

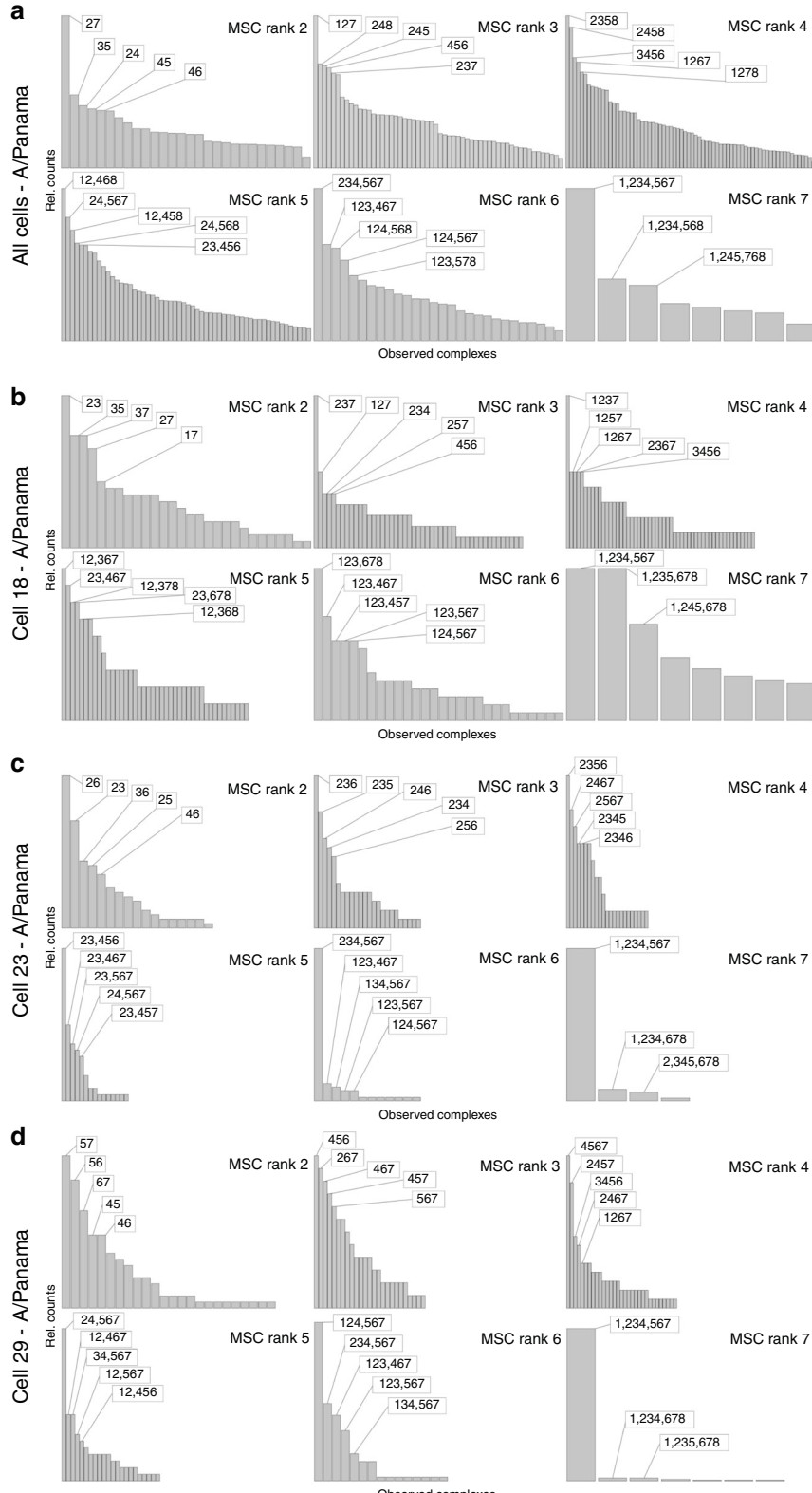

**Fig. 4 Specific segment combinations at intermediate MSC ranks after 10 h.p.i.** Each subplot shows the total relative counts of MSCs of each rank containing specific combinations of segments for all of the 69 analysed cells ('All Cells' **a**) and exemplary single cells ('Cells 18, 23 and 29' **b–d**, see also Fig. 3). Listed are the compositions of segments of the five most abundant complexes of each complex rank (boxes). To relate frequencies to the total counts of the MSCs of the specific rank we refer to Fig. 3 (left column). For better visualisation, data are normalised to the most abundant MSC of the respective rank. Source data are provided as a Source Data file.

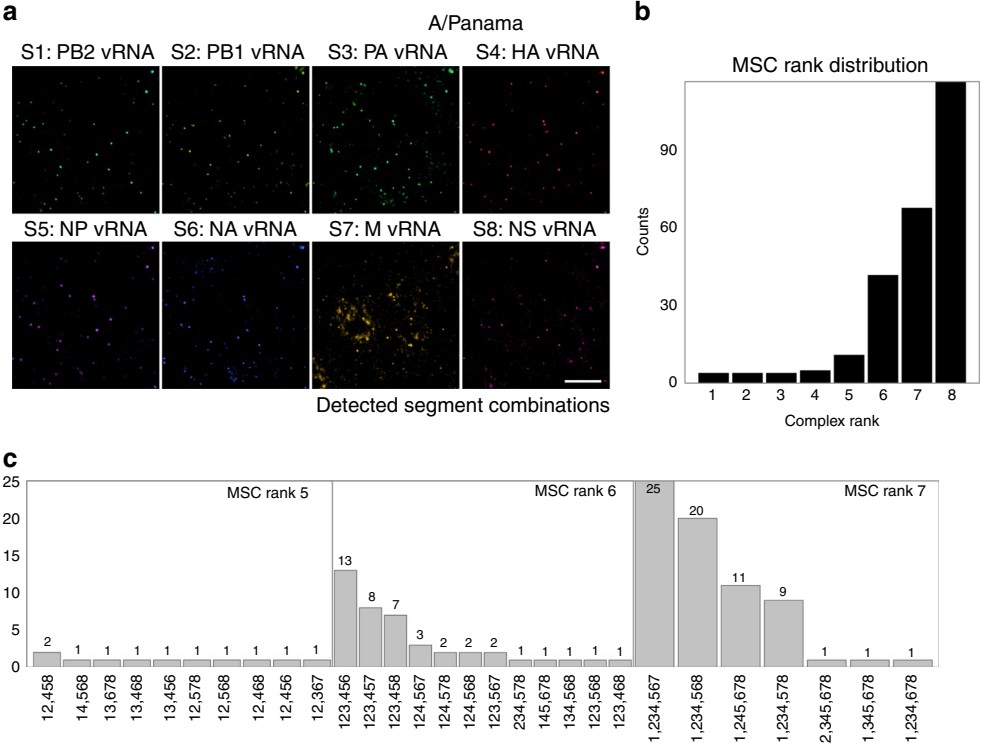

**Fig. 5 Viral ribonucleoproteins (vRNP) composition of intact A/Panama virions.** Virions were bound to A549 cells, and subsequently, virus-cell complexes were fixated. Viral RNA (vRNA) segments (S) were stained by fluorescence in situ hybridisation (FISH) with Atto550-coupled FISH probes. **a** Fluorescence images represent max z projections. Colours are identical to Fig. 2. Data were obtained from three independent experiments. Scale bar corresponds to 10 µm. **b** Histogram of all detected spots and their respective MSC ranks, same formatting as Fig. 3, black bars. **c** MSC distribution for ranks 5–7, same formatting as Fig. 4. Bars represent the number of observed MSCs of the respective combination. $N = 1653$ spots. Source data are provided as a Source Data file.

human A549 cells on ice followed by immediate chemical fixation. Interestingly, the frequency distribution of MSCs in A/Panama virions was clearly different from that found in A/Panama-infected cells being shifted to MSCs of high rank (Fig. 5). Indeed, MSCs of rank 8 composed of the eight different vRNP segments represented the most abundant fraction (46%), followed by rank 7 (27%) and rank 6 (17%). This finding supports previous observations implicating a mechanism that ensures a preferential incorporation of MSCs of high ranks into virions. Nakatsu et al.[39] applied scanning electron microscopy tomography to diverse influenza A and B strains and showed that at least 80% and in some cases even 100% of virions contained a complete genome arranged in the '7 + 1' pattern. Reverse genetics approaches have shown that deletions or mutations in a packaging signal sequence of a single vRNA can prevent the incorporation of other vRNPs into virions or virus-like particles[2,14,40]. Very likely, the dense arrangement of vRNPs in high-rank MSCs forms a surface favouring recognition and preferential incorporation of such complexes into assembling viruses. However, in contrast to Nakatsu et al.[39], our results show that incorporation of complete 8-rank MSCs into A/Panama virions is not perfect at all, reaching only 46%. We surmise that the fraction of virions with a complete set may vary depending on the strain of IAV[41] and, perhaps, the type of host cell involved.

**MSC assembly under non-permissive IAV infection.** All of the experiments described above refer to a permissive infection of human A549 cells by the human A/Panama virus of the H3N2 subtype[42]. To obtain insights into MSC formation in a non-permissive infection, we infected A549 cells with A/Mallard/ 439/2004 (Supplementary Fig. 10), a low-pathogenic avian IAV of the same subtype that we previously showed to cause abortive infections in these hosts[42]. Interestingly, the analysis of 54 cells revealed a heterogeneity of the relative frequency of segments and complexes in individual cells (Fig. 6a, b, see also legends for statistics). A more striking observation was the strong disruption of MSC formation, particularly high-ranking ones (Fig. 6c). The left-hand-sided, monotonically decreasing frequency distribution pattern of MSC ranks for the avian virus sharply contrasted with the u-shaped distribution we observed in permissive infections. The abundance of MSCs of rank 3 and higher was surprisingly low pointing to severe perturbation of vRNP bundling already in the early phase. A possible reason for the left-hand-sided frequency distribution pattern could be the low presence of segments being crucial for MSC assembly (Fig. 6a–c).

The reason for this observation needs future clarification. We have recently shown that replication of the avian A/Mallard/439/ 2004 virus in A549 cells is attenuated by three orders of magnitude at 72 h.p.i. in comparison to A/Panama/2007/99[42,43]. We were able to demonstrate that both viruses enter A549 cells with similar efficiency, and that mRNA from both virus strains accesses the translational machinery with comparable efficiency. Interestingly, A/Mallard infection in human cells was characterised by reduced M1 production and impaired nuclear export of NP, which may affect the nuclear export of and interactions between vRNPs. Hence, we consider it possible that defects observed for avian vRNP bundling in human cells are caused by disturbances in the ratios of available segment species in the cytosol. However, also in this case, further investigations are necessary, including more comprehensive analyses of bundling of avian IAV vRNPs in human and avian cells across a number of

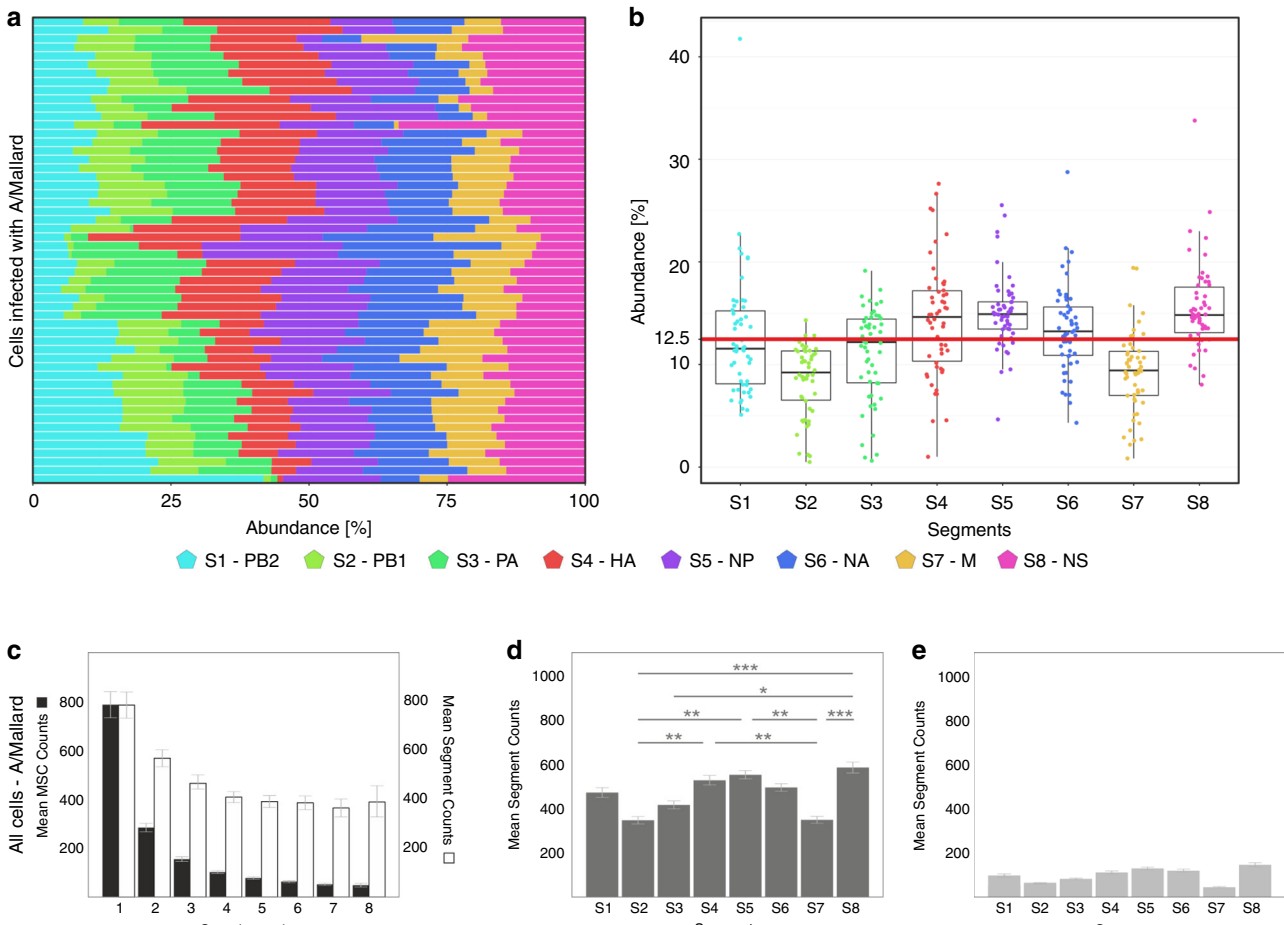

**Fig. 6 Viral RNA (vRNA) and MSC rank abundances in A/Mallard-infected A549 cells 10 h.p.i. a** Relative abundances of all vRNA segments per cell as measured by fluorescence in situ hybridisation (FISH). For each of the measured cells ($n = 54$ from three independent experiments), the number of spots that were detected was counted and normalised to the total number of spots per cell. The colouring scheme is the same as in Fig. 1. **b** Data shown in **a** in a Tukey boxplot representation. Upper bound of box, centre and lower bound of box represent the 75th percentile, the 50th percentile (median) and the 25th percentile, respectively. Upper and lower whiskers represent the maxima and minima of the boxplots showing the respective largest or smallest value within 1.5× interquartile range above the 75th or below the 25th percentile (bounds of box). In all, 12.5% (red line) corresponds to the expected fraction for each vRNA in a uniform distribution. For the segments, no significant difference (ANOVA, $F$ test, one-sided, $\alpha = 0.05$) between the cells was found ($P > 0.999$). The Bartlett test (one-sided, $\alpha = 0.05$) shows that the variability of the individual segments is significantly different ($P > 0.0001$) (see also legend to Fig. 3). The Kruskal–Wallis test does not presuppose homogeneity of variability, but agrees with the $F$ test in the result, i.e., no significant differences (one-sided, $P > 0.999$). Source data are provided as a Source Data file. **c–e** A total of $2 \times 10^5$ distinct segment spots were detected in A/Mallard-infected A549 cells and binned into about $9 \times 10^4$ MSCs. The frequency distributions of MSC ranks (**c** (filled bars: 'MSC counts', empty bars: 'Segment counts')), of all segments S1–S8 occurring in cells **d** and of free segments, i.e., rank 1, **e** are shown. Values are shown as mean ± s.e.m. Multiple comparison of means by Tukey test (one-sided, $\alpha = 0.05$) reveals the following significant differences between segments **d**: S2–S4, S2–S5, S2–S8, S3–S8, S4–S7, S5–S7 and S7–S8 ($P$ between 0.0001 and 0.0137; *$P \leq 0.01$; **$P \leq 0.001$; ***$P \leq 0.0001$). Source data are provided as a Source Data file.

different time points post infection for a validated conclusion. Furthermore, our recent results may also indicate that regular MSC assembly may not only depend on vRNP–vRNP interactions, but is co-regulated by host-cell factors. We showed in a comparative quantitative proteomics approach a differential regulation of a number of host proteins in response to A/Panama versus A/Mallard infections in A549 cells[42]. However, putative roles for these factors in the genesis and packaging of vRNPs remain to be investigated.

## Reliability of colocalisation analysis.

We are aware that colocalisation of segments and their assignment to the same MSC could in some instances be compromised by the optical resolution limit. In order to validate the analysis and respective conclusions on the composition of MSCs, we conducted several control

experiments. First, as a negative control of the algorithm, colocalisation of spots of two superimposed copies of the same image was analysed, whereby these two copies were rotated by 90° against each other. As expected, colocalisation was very low. The highest rank of MSCs observed was 2, with less than 1% of the spots of one image colocalised with spots of the other image rotated by 90°. Secondly, we investigated colocalisation of vmRNA species applying the same parameters used for analysis of vRNP colocalisation. Most frequently, we observed monomers and dimers (Supplementary Fig. 11), but complexes of higher ranks were rare. The fact that vRNAs and vmRNAs exhibited different patterns of (co)localisation showed that probes are specific for the respective RNA species, and suggests that the image analysis correctly assigned spots to complexes. Thirdly, using the centre of mass for each MSC, the distance of each spot from the centre of mass was calculated for x- and y coordinates

and plotted in a histogram (Supplemetary Note 2, Supplementary Equation 4 and Supplementary Fig. 12). Most segments showed a distribution with a marked peak relatively close to the centre of mass (25–60-nm distance). However, distances were still within the cylinder radius (300 nm, see above) that allowed colocalisation to be detected. Finally, to verify the confocal image-based MSC analysis further, we carried out super-resolution multi-colour STED microscopy (STED)[44] with a resolution of 79 nm for a selected set of segments (Supplementary Fig. 13). The major fraction of spots colocalising with other spots as determined by confocal microscopy was also colocalising at the resolution level of STED. Finally, as shown above, the frequency distribution of MSC ranks in infected A549 cells was very different between A/Panama and the low-pathogenic avian A/Mallard virus.

## Discussion

To fully understand the intracellular genesis of a complete octameric IAV genome, it would be preferable to track the formation of each complex and its integration into a budding virus over time. To date, we are not aware of any method that allows such a study in living cells. Our approach may offer a way out by the following argument. The analysis of IAV-infected human cells at an intermediate–late time point of infection (10 h.p.i.) revealed that all ranks of intermediate, not yet fully bundled genome complexes, were found. Hence, we suggest that the formation of viral genome complexes was in a steady state, at least still in progress and not completed, allowing us to characterise this essential dynamic process by means of microscopy on fixated cells.

The segmentation of the octameric IAV genome theoretically allows the existence of 255 distinct MSCs linked through a total number of 3025 assembly interactions, if one rules out that an MSC contains more than one copy of each segment. However, the total number of possible segment combinations will be considerably smaller, if assembly is dictated by specific rules, i.e., specific interactions that favour specific bundling pathways.

Overall, our results implicate the existence of selective packaging rules with limited flexibility by which vRNPs are assembled into a complete IAV genome. To identify potential routes for the bundling of segments, we analysed the most frequently observed MSCs within each individual rank (see Fig. 4a, 'All Cells'). We explored the intuitive notion that the detection of prevalent MSC compositions at different ranks provides evidence for the existence of common packaging pathways in cells. It turned out that in most cases, these complexes could be connected consistently assuming a sequence of monomeric addition reactions (Fig. 7).

Several possible continuous pathways can be observed in the data, another strong indication for a flexible selective packaging that can occur through several alternative routes. The longer segments as S1, S2 and S4 occur already frequently in MSCs of lower ranks, which is a hint for a relevant role of longer segments in MSC assembly as suggested by Noda et al. and Fournier

et al.[11,12]. Notably, in the pathways shown in Fig. 7, segment S8 is integrated into MSCs preferentially at the later steps of MSC assembly. This suggests that S8 may not be essential for initiation or early steps of vRNP bundling (see also below).

While we cannot rule out that some of the MSCs we identified represent dead ends of genome assembly, we have several indications to suggest that MSCs with prevalent compositions at intermediate ranks reflect essentially complexes en route towards fully assembled MSCs of rank 8. If the MSC distributions were dominated by irreversible dead ends, we would very likely observe an enrichment of such MSCs in the intermediate ranks. We found, however, reduced numbers of intermediate MSCs, as it is obvious from the overall u-shaped distribution of the total MSCs in cells (Fig. 3a, b). The clear majority of segments were organised in MSCs of high ranks (Fig. 3a, b). The preferential occurrence of completely packaged octameric MSCs with respect to MSCs of intermediate rank cannot be explained by purely random colocalisation. Secondly, highly abundant segment combinations of one MSC rank usually had counterparts in higher MSC ranks containing the same segments (such as 2–4, 2–4–8, 2–4–5–8, 1–2–4–5–8, …; Fig. 4a 'All Cells'), which indicate successive steps of assembly. As shown in Fig. 7, the very frequent MSCs in the various ranks can be continuously connected with a certain degree of flexibility in ascending order from rank 1 to 8. Of note, although we assumed the formation of the complete genome on the basis of the association of a free segment with an existing complex (Fig. 7), future studies should address whether genome assembly could also proceed by association of MSCs of rank 2 or, perhaps, even higher.

To support the existence of a packaging model that is mainly affected by vRNA interactions and is consistent across cells, we constructed a mathematical model under the assumption that MSCs of rank $k$ ($k \in [2,7]$) can only be formed by non-reversible addition of a free segment to MSCs of rank $k$–1. We used a fully parameterised regression-based model that assumes a (possibly) different rate constant for each association of a free segment to an MSC (see 'Methods'). The performance of the model was evaluated by training the parameters on subsets of the data (training data), and computing the predictions of the model for the unseen data (test data) with a fivefold cross-validation scheme (see 'Methods'). The model was able to predict the abundances of individual MSCs for single cells with high accuracy (MSE = 37.27, $P$ value for outperforming a random model <0.05) (Supplementary Fig. 14 and Supplementary Table 2). These results support the existence of a set of 'rules' for the packaging pathways, which are an implicit function of the abundances of free segments that are available. We additionally confirmed that prevalent types of MSCs form continuous pathways in single cells: MSCs of rank $k$ ($k \in [2,7]$) are significantly enriched for pathways that originated from the most abundant MSCs of rank $k$–1, and are followed by the most abundant MSCs of rank $k + 1$ ($P$ value < 0.05).

Of particular interest for understanding the potential role of distinct segments in MSC assembly is the analysis of MSC composition in those cells in which one vRNP species is of low abundance or even absent. Very likely, at high MOI as used in our experiment, such a situation would occur only very rarely in agreement with our observation. However, we found a few cases as shown for 'Cell 29' and 'Cell 23' (Fig. 3g, h) with a strongly significantly reduced copy number of S8 and of S1 and S8, respectively.

These two segments were only identified in MSCs of higher ranks (Fig. 4c, d). As explained in detail in Supplementary Discussion, the analysis of MSC frequency distribution of 'Cell 29' indicates that S8 is dispensable in early stages of the segment-bundling pathway, and is only added in the final steps of MSC formation of rank 8. Analysis of 'Cell 23' points towards an

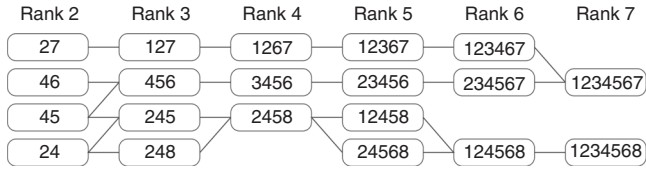

**Fig. 7 Potential bundling routes in A/Panama-infected A549 cells.** The most abundant MSCs are those represented in Fig. 4 ('All Cells'). Joining lines were drawn if two MSCs of adjacent ranks with a specific composition of segments could be connected to the next through the addition of a single segment. The numbers in the boxes correspond to segments S1–S8.

important role of the long vRNP segment S1 already in formation of MSCs of lower ranks. These conclusions must be taken with caution as only a few of these cells were observed. Nevertheless, these examples illustrate that frequency distribution analysis can be a very useful tool to uncover the role of distinct vRNPs in MSC assembly (see 'Supplementary Discussion').

As RNA–RNA interactions are supposed to play a pivotal role in bundling of vRNP segments, it is of interest to address if our results on preferred vRNP compositions of intermediate MSCs are in agreement with intermolecular interactions between vRNAs that have been recently obtained by analysis of cross-linked vRNAs (Dadonaite et al.[37]). This study has shown for H1N1 (A/WSN/1933 and A/PR8/34) and H3N2 (A/Udorn/307/72) viruses that the network of interactions between closely related viruses is broadly similar, but there can be significant differences between more distant viruses. Since we investigated a H3N2 strain (A/Panama), we can compare the results of Dadonaite et al.[37] regarding H3N2 with ours. They found for S8 only a small number of interactions with other segments for the H3N2 virus (see Fig. 3 in Dadonaite et al.[37]). In contrast, there were numerous interactions of segments S1–S3 among each other and with other segments. Our findings on the role of S1 and S8 in MSC formation are consistent with their observations, indicating a prominent role in the early phase of MSC formation of S1, but not of S8 (see Fig. 7).

Our findings suggest that IAV genome packaging proceeds in a flexible and selective manner along a number of alternative bundling pathways supporting the view that vRNPs form patterns more complex than one-on-one interactions[11,12]. This conclusion is supported by Dadonaite et al.[37] revealing the existence of multiple specific interactions between vRNP segments in a high-resolution structure of the IAV genome. The study suggested that even the same nucleotide stretch of a given vRNA sequence can interact with different other RNA segments, strongly implicating a significant redundancy and flexibility of the vRNP–vRNP interaction network while preserving the formation of the IAV genome complex with eight different vRNPs. vRNP bundling would be expected to proceed inefficiently when strictly only one specific gene segment could be added at a time to the growing MSC. However, maintaining some degree of redundancy within the genome-packaging network confers robustness to the bundling process if several alternative segments were allowed to assemble at each given step. Moreover, a flexible packaging mechanism could buffer against possible limitations caused by low expression levels of cellular components engaged in vRNP packaging, naturally found between single cells of a given type. Segmented RNA viruses, particularly IAV, undergo reassortment, an evolutionary mechanism that could take advantage of imperfect or at least flexible genome packaging[45].

## Methods

**Oligonucleotide design of FISH probe sets.** The eight vRNP segments code for the following major viral proteins: S1—PB2, S2—PB1, S3—PA, S4—HA, S5—NP, S6—NA, S7—M1 and M2 and S8—NS1 and NS2. The numbering of the segments is based on their lengths in terms of the total number of nucleotides, in descending order (i.e., PB2 has the longest sequence). Stellaris probe designer[46] was used to obtain a maximum number of oligonucleotides that specifically target influenza A/Panama/2007/1999 virus vRNAs and vmRNAs (Supplementary Table 1). In the design of M1 and NS1 mRNAs, only sequences not present in M2 or NS2 mRNAs were considered since the latter are splicing products from the vRNAs encoding M1 and NS1, respectively. The numbers of oligonucleotides complementary to the latter two RNAs did not suffice for the design of probe sets containing at least 15 distinct and non-overlapping 20-mer oligonucleotides. For this reason, M2 and NS2 mRNAs were not investigated. The oligonucleotide seeds that were obtained were blasted[47] against human RNA to further filter out those that might cause off-target hybridisations in cells. In an additional step, NUPACK[48] was used to predict the degree of secondary structure formation of the remaining oligonucleotide sequences to further exclude probes that were far less likely to hybridise. The first and last 30 nucleotides of each vRNA species were excluded due to their high

degree of conservation among all the segments. Oligonucleotides were purchased from Biomers.net (Ulm, Germany) with C6 aminolinkers at the 3′ ends. These nucleotide sequences were coupled to the succinimidylester-modified dyes Atto550 (ATTO TEC, Siegen Germany) and Abberior® STAR635P (Sigma-Aldrich, St. Louis, MS, USA), respectively, and purified by HPLC according to the manufacturer's protocol. The same protocol for the design and synthesis was applied for FISH probes to detect vRNAs of A/Mallard/439/2004.

**Virus, cell culture and infection.** The prototypic seasonal influenza A/Panama/2007/99 virus (H3N2) (NCBI accession numbers: DQ487333-DQ487340) was propagated and plaque-titered in MDCK type II cells (ECACC 00062107)[42]. To this aim, confluent MDCK cell layers were infected with A/Panama at MOI 0.01 and incubated at 37 °C for 2 days. In the case of successful infection, the cells detach from the surface within this time period, and virus particles are released into the medium. Cellular debris was removed from the virus suspension by centrifugation. To measure the plaque-forming units (PFU) of the suspension, MDCK cells were grown in six-well plates until full confluency was reached. The different wells were infected with 10× dilution steps of the virus suspension, respectively, and cells were immobilised afterwards with agarose overlay medium. After 2 days of incubation at 37 °C and 5% $CO_2$ agarose overlay medium was removed, cells were fixated by 10% formalin and PFU was determined. Stocks of the influenza viruses A/Mallard/439/2004 (H3N2) (GISAID accession numbers EPI859640-EPI859647) were grown in the allantoic cavities of 10-day-old embryonated chicken eggs for 3 days at 37 °C[42].

Human A549 lung epithelial cells (ATCC CCL-185) were cultured in Dulbecco's Modified Eagle Medium (DMEM (+), DMEM supplemented with 1% penicillin/streptomycin, 2 mM L-glutamine and 10% foetal bovine serum, Pan Biotech, Aidenbach, Germany) at 37 °C and 5% $CO_2$.

For experiments, cells were seeded in ibidi μ-Slides VI0.4 (ibidi, Munich, Germany) and were infected the next day with IAV at a multiplicity of infection (MOI) of 5, or mock-infected (without virus) as a control. While A/Panama causes a permissive infection of A549 cells, infection of these cells by A/Mallard is non-permissive[42]. To achieve synchronised infections, samples were washed with ice-cold DPBS++ (Pan Biotech, Aidenbach, Germany) and incubated with ice-cold infection medium (DMEM supplemented with 1% penicillin/streptomycin, 2 mM L-glutamine and 0.2% foetal bovine serum albumin) on ice for 20 min to allow adherence of viruses on the cellular surface. To initiate infection for late-infection experiments, samples were incubated at 37 °C and 5% $CO_2$ for 45 min, and subsequently were washed with warm DPBS++ and incubated with infection medium at 37 °C and 5% $CO_2$ until fixation at 10 h post infection (p.i.). To fixate cells, samples were washed twice with DPBS++ before and after incubation with 10% formaline (Sigma-Aldrich, St. Louis, MS, USA) at RT. Finally, ibidi slides were stored at 4 °C in 70% ethanol overnight for cellular permeabilisation and protection against RNase activity. For single-virion experiments, samples were fixated upon adherence to cells on ice for 20 min.

**Immunofluorescence and DAPI staining.** Immunofluorescence was performed prior to FISH cycles because the formamide-removal buffer has been reported to unfold proteins[49], and here was found to perturb antibody staining. To block an unspecific binding of antibodies, cells were treated with 0.2% acetylated BSA (B8894, Sigma-Aldrich, St. Louis, MS, USA) in 2×SSC buffer (saline–sodium citrate buffer, supplemented with 2 mM of the unspecific RNase inhibitor vanadyl ribonucleoside complex VRC, Sigma-Aldrich, St. Louis, MS, USA) at RT for 15 min before incubation with a FITC-conjugated anti-IAV-NP antibody (NP-FITC, Merck Millipore, Darmstadt, Germany) at a dilution of 1:500 in BSA-containing 2×SSC buffer at RT for 45 min. To remove unbound antibodies, samples were washed twice with 2×SSC buffer at RT for 10 min. Subsequently, cells were incubated with 100 nM DAPI at RT for 10 min to stain nuclei. Again, to remove unbound probes, the samples were washed twice with 2×SSC.

**Fluorescence in situ hybridisation.** To enhance FISH staining signals by increasing the accessibility of vRNA for probes, samples were initially treated with warm 80% formamide at 37 °C for 10 min (see above). This was followed by rehydration with 2×SSC buffer at RT for 10 min. Subsequently, cells were incubated with hybridisation buffer (200 nM FISH probes, 2×SSC, 10% formamide, 10% dextrane sulfate and 2 mM VRC) at 37 °C for 2–4 h, and were washed twice with warm 10% formamide in 2×SSC at 37 °C for 10 min. Two protocols were tested to determine the best procedure for removing FISH probes between sequential labelling steps: (i) cleavage of DNA-FISH probes by DNase I, and (ii) decrease of melting temperature of double-stranded nucleic acids by 80% formamide buffer. Only the latter yielded a suitable, multiple repeatable labelling of targets (Supplementary Figs. 2 and 3). Formamide treatment caused a significant enhancement in the fluorescence intensity of FISH staining (Supplementary Fig. 2), in particular after the first formamide treatment, while maintaining a constant FISH signal pattern and a very low background signal. Treatment with high concentrated formamide buffer impaired immunofluorescence staining of NP, however, so all MuSeq-FISH experiments were performed only after fluorescence imaging of antibody staining.

In consequence, subsequent to image acquisition, oligonucleotide probes were removed by washes with warm 80% formamide at 37 °C for 10–15 min followed by rehydration with 2×SSC at RT for 5 min. Removal of probes was verified by microscopy before initiation of a new FISH cycle. Each FISH staining cycle labelled two different target RNAs simultaneously. To enhance confidence in our colocalisation analysis, each vRNA was targeted twice in the course of FISH cycles by specific oligonucleotides bearing different attached fluorophores: one using Atto550-labelled and the second STAR635P-labelled oligonucleotides. In addition, the vRNA/vmRNA signals were stained in a random order that varied in each of the replicates. Because the vRNA and vmRNA from the same segment were complementary, we stained only RNAs originating from different segment species simultaneously; this prevented partially complementary probe sets from hybridising with each other.

**Confocal microscopy**. Images were acquired with a Visitron VisiScope scanning-disc confocal laser microscope (Visitron Systems, Puchheim, Germany) and a ×60/1.2 UPlanSApo water or a ×100/1.3 UPlanFLN oil objective (giving rise to a pixel size of 0.13 and 0.2 μm, respectively). An Andor iXon 888 EMCCD camera (1024 × 1024 pixels, Andor, Belfast, Northern Ireland) was used to detect fluorescence. Excitation was carried out using the following diode lasers: 488 nm (FITC) with an ET525/50-nm emission filter, 561 nm (Atto550) with an ET600/50-nm emission filter, 640 nm (PCA635P) with an ET700/75-nm emission filter and 405 nm (DAPI) with an ET460/50-nm emission filter. Each image was captured with a resolution of ∼200 × 200 × 700 nm and with a 0.40-μm z-step size with 26–29 slices spanning the entire cell volume.

**Image preprocessing—alignment and spot detection**. Sequential imaging requires the physical removal of the specimen from the microscope, so any offset that occurred in the raw images was corrected in the *xy* plane using the 'Template Matching and Slice Alignment' plugin in ImageJ[50], while the z direction was aligned manually.

3D Spot detection was performed on raw images using FISH-quant[51]. The threshold was set to ensure minimal hits for control samples, i.e., for non-infected cells. While multiple formamide treatments increased the fluorescence intensity (see 'Results'), this did not affect spot detection because FISH-quant only distinguishes the signal intensities of FISH spots from background fluorescence by a fixed threshold value without further evaluation. The borders of individual cells and their nuclei were identified by taking FISH staining for cellular volume and DAPI staining for nuclei using the built-in cell outline tool (nuclei were excluded from the analysis). The general workflow was performed according to the FISH-quant tutorial. In single-virion experiments, spot identification was done manually in FIJI[52,53].

**Colocalisation analysis**. The vRNA spots that were identified were analysed for colocalisation using a custom-written R script (R version 3.2.4, available via GIT-HUB (https://github.com/Budding-virus/Packbund). A brief summary of the steps performed by the algorithm includes: (1) loading all images from a particular imaging position, (2) correcting *x/y/z* offsets, (3) discarding all spots that did not appear in both detection channels (Atto550/STAR635P), (4) serial colocalisation analysis, placing all spots within a cylinder (radius = 300 nm, height = 1000 nm) into one MSC (colocalisation must fit all criteria—i.e., segments being within a cylinder of 300 nm *xy* and 1000 nm *z*), (5) computing the centre of mass for each MSC and (6) discarding all MSCs with multiple copies of segments within one complex (less than 5% of MSCs, see 'Results').

**Mathematical model**. We trained a regression-based model (available via GIT-HUB (https://github.com/Budding-virus/Packbund) for the linear relationship between the abundances of MSCs of rank $k$ ($k \in [2,7]$) and the products of the abundances of their constituent segments and the corresponding rank $k–1$ MSCs. For example, the MSC composed of segments Sx, Sy, Sz could potentially be assembled in three ways: from segment Sx joining the MSC composed of Sy and Sz, from segment Sy joining the MSC composed of Sx and Sz or from segment Sz joining the MSC composed of Sx and Sy. The performance of the model was evaluated by partitioning the data (the abundances of all MSCs across all single cells) into $k$ ($k = 5$) subsamples in which the parameters of the subsamples are learned on $k–1$ subsamples, and the predictions of the model are compared to the experimental data (which was withheld from training) for the remaining sub-sample, and are averaged over the results for all k-possible partitions (*k*-fold cross-validation). The MSE (mean-squared error) calculated for the predictions of the model for all 246 ranks $k$ ($k \in [2,7]$) MSCs relative to the levels of the corresponding MSCs measured in the cells was found to be 37.27 ($P$ value < 0.05). This measure was assigned a $P$ value based on a standard permutation test (where the levels of the rank $k$ MSCs being predicted were permuted). The correlation coefficient of the predicted abundances and the corresponding experimentally measured abundances was found to be 0.88 ($P$ value < 0.05).

**STED microscopy**. The measurements were performed on a custom-built STED setup[44], in which a Gaussian-shaped excitation beam was superimposed to a doughnut-shaped STED beam to yield lateral resolutions below the diffraction limit. The excitation light pulses of 584-nm and 652-nm wavelength were extracted from the emission of a supercontinuum laser source (EXWB-6, NKT Photonics, Birkerød, Denmark) using an acousto-optical beam splitter. For STED imaging, the targeted de-excitation was performed by laser pulses of 775-nm wavelengths (Katana08-HP, Onefive, Zurich, Switzerland). The STED laser was set to deliver 40 mW at the back focal plane of the objective. Both lasers were synchronised at a repetition rate of 40 MHz. A quad scanner[54] was used to scan the beam pair laterally across the sample. Axial scanning was done by moving the objective with a piezo element. Fluorescence from the sample was detected in four spectral channels with a common confocal pinhole for intrinsic, stable colocalisation of the detection volumes. Each channel features an avalanche photodiode (APD) as a detector. The spectral position and width of each individual channel were defined by commercial long-pass filters[44]. Here we used four detection channels ranging from 605 to 642 nm, 642 to 675 nm with a gap from 646 to 671 nm to block the excitation wavelength of 652 nm, 675–700 nm and 700–745 nm. Individual images of the z stack were recorded with interleaved line steps, during which the excitation wavelength was alternated. Each line was scanned twice with a pixel dwell time of 10 μs for confocal imaging and with a pixel dwell time of 120 μs each for STED imaging. The pixel size was 25 nm by 25 nm in *xy*; individual z slices were spaced 250 nm apart. The resolution of confocal and STED images was evaluated on a sample of dispersed fluorescent beads of 20-nm diameter (Life Technologies, crimson beads, cat. #F8782), which were immobilised on a poly-L-lysine-coated coverslip (Sigma-Aldrich). The individual full width at half maxima (FWHM) were measured to be 300 nm laterally and 710 nm axially in the confocal mode, and 79 nm laterally and 830 nm axially in STED mode.

Three separate rounds of infection and staining were performed. Each time, mock-infected cells were stained as control to check target specificity. Single-colour stainings of the same vRNA targets were performed to obtain the intensity distribution of each dye in the individual detection channels for spectral unmixing. Data were acquired with a field of view of 8 μm by 8 μm. Every data stack was recorded at a different position in the sample, i.e., different cell, to prevent false results due to variations in the sample. At least three images were acquired for control stainings, and at least five images for three- and four-colour stainings. All raw images were manually scanned for the presence of all fluorescent labels and for a good signal-to-noise ratio. Usually three out of five images showed high indication of all fluorophores with satisfactory signal levels. These were consequently unmixed with the non-negative matrix factorisation algorithm used by the Spectral Unmixing Plugin for ImageJ from Joachim Walter, which is based on the non-negative matrix factorisation algorithm by Neher and Neher[55]. The spectral distribution of the emission of single-fluorophore species was determined from reference images in which the same vRNA targets were labelled. After unmixing, the z stack of each colour channel was smoothed in 3D with a Gaussian with a width of 0.5 px in each direction in ImageJ for better visualisation.

**Reporting summary**. Further information on research design is available in the Nature Research Reporting Summary linked to this article.

## Data availability
Source data underlying Figs. 2a, b, 3a–l, 4a–d, 5b, c, 6a–e and Supplementary Figs. 3b, 8, 9, 11 and 14, and for MSC per cell for A/Panama and A/Mallard, and segment distributions (also incl. data for Supplementary Fig. 7), are provided with this paper as a Source Data File. Source data are provided with this paper.

## Code availability
Codes for the custom-written R script for colocalisation and for the regression-based model for the linear relationship between the abundances of MSCs of rank k and the products of the abundances of their constituent segments and the corresponding rank $k–1$ MSCs are available via GIT-HUB (https://github.com/Budding-virus/Packbund). Source data are provided with this paper.

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

## Acknowledgements

For helpful discussions, we thank Anne Sadewasser (Robert Koch Institut Berlin) and Edda Klipp and Max Schelker (Humboldt-Universität zu Berlin). This work was supported by the German Ministry of Education and Research (0316180 and 0316180D, eBio: ViroSign to A.H. and T.W.), the Einstein Foundation (A-2012-140, single-molecule RNA to A.H. and O.S.), the German-Israeli Helmholtz Research School SignGene (to S.P.), the Deutsche Forschungsgemeinschaft (TransRegio 84 project B2 to T.W.) and the Leibniz Graduate School (to M.S.). We acknowledge support by the Open Access Publication Fund of Humboldt-Universität zu Berlin.

## Author contributions

I.H., S.P. and M. Schade designed the project; I.H., S.P.M. Schade and N.N. performed and analysed FISH experiments; M.N., N.F. and M. Schreiber modelled the data; M.L.-K. and F.W. performed and analysed STED experiments; K.J. and F.J. performed qRT-PCR experiments; J.C. and O.S. labelled probes; T.W. provided virus samples; A.H. and T.W. supervised the project; I.H., S.P., T.W. and A.H. wrote the paper with input from all authors.

## Funding

## Competing interests

The authors declare no competing interests.
