## [Peer Review File · Nature Communications]

Reviewers' Comments:

Reviewer #1:

Remarks to the Author:

In the manuscript entitled, "Selective, flexible packaging pathways of the segmented genome of Influenza A Virus", the authors utilized sequential FISH staining to visualize all eight influenza viral segments within a single infected cell at a single timepoint during infection. In this approach, the authors used two dyes per vRNA segment in random order to validate their staining. Importantly, the authors imaged each segment twice with FISH probe sets conjugated to two different fluorophores to isolate the reproducible intracellular localization of each segment. The overall conclusions of this study are that, 1) the most abundant vRNP complexes are those with either 1 segment (rank 1) or 8 segments (rank 8) while Intermediate ranks are found at lower abundances, 2) higher ranked complexes are found more often in intact virions, 3) complexes rarely contain multiple copies of the same segment, and 4) disruption of multi-segment complexes (MSC) in human cells infected with an avian virus.

Uncovering the mechanism by which IAV genome segments assemble is crucial to understanding reassortment and virion production. Visualizing these vRNA segments by FISH, especially all 8, is a powerful tool to study influenza selective assembly. The data presented supports the current model of IAV genome assembly where vRNP segments form bundles, the study also supports a newer model that the vRNP interaction clusters are flexible in nature as demonstrated with RNA-RNA interaction mapping in purified virions (PMID: 31332385). However, the study presented here has the advantage of knowing the precise three-dimensional location and composition of each vRNP bundle within a cell yet they do not utilize this information to further refine the vRNP networks at a cellular level. Given the recent data demonstrating that heterogeneity in viral infection at a single cell level (PMID 29451492 and PMID 30181264), the authors should consider whether the multiple bundling routes are a consequence of including cells that do not contain all 8 segments into their analysis. Overall the visualization of all 8 segments within infected cells and in bound virions is unique and this technique serves as an additional strategy to confirm a model of influenza assembly where the vRNP form subcomplexes that are flexible as demonstrated by biochemical and recent machine learning approaches (PMID: 30689627).

Overall, the clarity of the study could be improved with attention to the following issues.

1. The authors should provide experimental validation of the specificity for each probe set against other segments as well as any cRNA/vmRNA. Precise probe specificity is critical for accurate conclusions to be drawn from these data sets. It is also important to note that the authors state that the colocalization algorithm was a custom script but have not publicly shared the code or imaging data sets.
2. How do the networks defined by the authors in Fig 6 compare to the recently published RNA-RNA interactions for the Udorn or Wyoming/03/2003 H3N2 virus in PMID: 31332385. In particular, the lack of segments 7 and 8 in any of the initial complexes is intriguing and it may be due to the use of "all cells" in this analysis rather than cells that had an equal proportion of all 8 segments. Based on the data in Fig 3, segment 8 was under represented more often compared to the other segments and that may bias the networks constructed in Fig 6.
3. On page 8, the authors stated that, "we never obtained evidence for more than two copies of the same segment in an MSC; less than 5% of the MSCs contained two copies of the same segment". However, volume binning for vRNA spots in this study was 300 x 1000 nm, when they reference that a vRNP is 50-150 nm. If this statement was derived from the FISH experiments, it would be impossible that single vRNA of a specific segment could be differentiated given the limits of their methods and imaging resolution of standard confocal microscopy. This conclusion should be experimentally validated using approaches similar to PMID: 22547828 or else do not make such a claim.
4. The current model of IAV MSC assembly suggests that segments assemble as MSCs translocate from the nuclei to the plasma membrane. If so, lower ranked complexes would be observed near the

nucleus with higher ranked complexes near the plasma membrane. The authors should analyze ranks of complexes within various distances from the nuclei and/or plasma membrane.

5. In Fig. 5C, it appears that the frequency of MSCs containing S1 is greater than S2 and so on (frequency of $S1 > S2 > S3 > \dots > S8$). In virions with MSCs ranked less than 8, is the probability of which vRNP segments make it into these particles linked to segment size? In this study, are larger segments found more often in lower ranked MSCs compared to smaller segments during infection? This could be an artifact of the number of probes capable of binding to a given segment and should be acknowledged in the text.

6. In this study, the authors used a low-pathogenic avian IAV, A/Mallard, to study MSC assembly under non-permissive IAV infections. Given that infection with this virus is abortive in human A549 cells, the rationale for this experimental design is unclear. The replication kinetics of A/Mallard versus A/Panama are likely different and thus a 10 hpi cross-section between these infections may not be comparable. The complexity of this comparison makes these data difficult to interpret why the pattern of vRNP colocalization are different between these two virus infections.

Reviewer #2:

Remarks to the Author:

Haralampiev, Prisner, et al. used a multiplexed FISH assay to investigate all 8 influenza vRNA segments in parallel within A549 cells infected with human or avian influenza A virus. These data support the prevailing model of influenza virus genome packaging in which each one of eight-influenza virus vRNPs are selectively bundled into multi-segment complexes and then subsequently incorporated into newly forming virions.

The authors developed significant improvements to an existing approach in order to answer an important and interesting question pertinent to the influenza field with relevance to other viruses with segmented genomes. The data have the potential to advance current understanding of influenza A virus genome assembly, an area of high interest in the field. However, a more rigorous analysis and systematic presentation of the data is greatly needed. The current version of the manuscript is unclear and poorly organized, to the extent that it is difficult to fully evaluate the results. The authors should consider rectifying the following major and minor concerns:

1. The introduction would benefit from a more comprehensive summary regarding several topics: i) if you are going to present reassortment as an evolutionary benefit of genome segmentation, you should also mention fitness costs of reassortment ii) packaging model—if you are going to mention both random vs selective packaging models, you should present evidence for the prevailing model (selective) but note the limitations of the previous studies, iii) expand on the limitations of current understanding of influenza virus genome packaging mechanism and the signals that direct packaging.
2. Quantification of observed differences were included throughout the manuscript. For example, avoid vague phrasing such as “mostly a rather balanced ratio”, “approximately equal”, “slightly lower than average” and “highly similar” and quantify these differences using statistics.
3. In sub-section 2 of the results section, expand on the rationale for determining vRNA spot dimensions and exclusion of nuclear spots. Also reference data to support the statement that <5% of MSCs showed 2 copies of the same segment.
4. An important aspect of this study is the inclusion of human vs avian influenza A strains infecting a human cell line. The results would be more meaningful if presented in a way that cohesively and linearly presents these data (i.e., the figures should be presented chronologically in the text, e.g., figure 2C/D should be interpreted before analyzing Figure 3) with emphasis on comparing/contrasting human vs avian influenza A-infected cells.
5. Figure 3. Presentation of the data that shows the variance among the different cells (with statistical

analysis) is needed. It is unclear whether row 1 represents an average over all cells (as mentioned in the text) or total number of MSCs or segments. Showing averages, with bars including error bars, would allow the authors to make meaningful comparisons between ranks rather than referring to the u-shaped distribution

6. Fig. 4. Rather than showing segment counts for each MSC for each cell, it would be more impactful to consolidate these data to show major trends (or lack thereof) to more clearly articulate the conclusion that there is no dominant MSC composition within each rank.

7. 'Reliability of colocalization analysis' sub-section of the results should be included earlier on in the text. These are important controls that should be presented before the data, but do not necessarily need a separate section and can be integrated throughout the text.

8. 'Implications for IAV genome assembly' sub-section should be moved to the Discussion/Conclusions section of the manuscript.

9. Please include panel letters on all figures for clarity throughout the figure legends and text.

10. Figure legends should not contain results or interpretation of the data shown, except for a descriptive title if warranted.

11. Include more descriptive y axis titles for Figure 2A/C (e.g., Cells infected with A/Panama or Cells infected with A/Mallard).

12. Include axis labels for Fig. 5C. If represented on the same graph, scale of y-axis needs to be consistent. Purple color for NP vRNA is not distinguishable in image in Panel A.

Reviewer #3:

Remarks to the Author:

The paper by Haralampiev et al. investigates the bundling of the 8 influenza A virus (IAV) genome segments by a multiplexed FISH assay. The authors could provide first insight into the cytoplasmic segment composition and possible genome complex intermediates.

The paper describes for the first time a robust FISH-based assay to study genome bundling of all IAV segments and thus provides an excellent basis for future studies. However, besides developing a robust FISH assays, the authors did not really apply their method to address some open questions in the field of genome packaging.

Major points:

1) To convincingly demonstrate that the avian IAV used in this study is able to bundle correctly the authors should study the vRNP composition also in avian cells.

2) Packaging sequences in the viral genome are required for correct packaging of the 8 different genome segments into a virus particle. It is believed that these sequences orchestrate genome bundling. Thus, it is not clear why the authors did not investigate a packaging mutant virus. This would provide valuable information.

3) There are published data available suggesting that vRNP bundling increases en route to the plasma membrane. This information is lacking and if technical possible should be included.

4) There are Panama virus-infected cells where 8 bundles are not efficiently formed, because some segments are poorly expressed. This suggests that balanced genome expression levels are important. To address this point experimentally the authors might consider to generate a virus that expresses less efficiently only one segment.

Minor points:

- 1) Control experiments (page 15: Reliability....) is very difficult to understand.
- 2) In the supplement all analyzed cells (Panama and Mallard) can be shown.
- 3) Show pictures of genome distribution in Mallard-infected cells
- 4) Infection of human cells with the Mallard virus: are semi-infectious virus particles released from the cells?
- 5) Is the introduction of the human PB2 signature in the Mallard strain sufficient to restore genome bundling?
- 6) ...mediating interactions between vRNPs (10,13-23)Ref <https://doi.org/10.1073/pnas.0437772100> is missing.
- 7) ...Fluorescence in situ hybridization (FISH) studies (24-26). Ref 24 is not accurate because that was FISH in particles.
- 8) Definition of packaging in the footnote: For the reviewer packaging is the incorporation of the genomes into viral particles.
- 9) ...In fact, considering all possible combinations of vRNPs, the random packaging model predicts that 98% of the MSCs would contain two or more copies of a distinct segment species: How do the authors come to 98%?
- 10) ... To assess the vRNP composition of mature A/Panama virions by MuSeq-FISH, virions present in virus stocks were allowed to bind to the surface of human A549 cells on ice followed by immediate chemical fixation. Interestingly, the frequency distribution of MSCs in A/Panama virions was clearly different to that found in A/Panama-infected cells being shifted to MSCs of high rank (Fig. 5): How valid is the comparison? Which virus stock was used? One after infection with a high MOI or one after infection with a low MOI?
- 11) The authors analyzed only one time point post infection and one infection dose. Is the vRNP composition different at earlier time points and at lower MOIs?

We would like to thank the reviewers for the encouraging evaluation of our manuscript and their valuable comments and additions. In the following we will address all points in detail. Reviewers' comments:

Reviewer #1 (Remarks to the Author):

In the manuscript entitled, "Selective, flexible packaging pathways of the segmented genome of Influenza A Virus", the authors utilized sequential FISH staining to visualize all eight influenza viral segments within a single infected cell at a single timepoint during infection. In this approach, the authors used two dyes per vRNA segment in random order to validate their staining. Importantly, the authors imaged each segment twice with FISH probe sets conjugated to two different fluorophores to isolate the reproducible intracellular localization of each segment. The overall conclusions of this study are that, 1) the most abundant vRNP complexes are those with either 1 segment (rank 1) or 8 segments (rank 8) while Intermediate ranks are found at lower abundances, 2) higher ranked complexes are found more often in intact virions, 3) complexes rarely contain multiple copies of the same segment, and 4) disruption of multi-segment complexes (MSC) in human cells infected with avian virus. Uncovering the mechanism by which IAV genome segments assemble is crucial to understanding reassortment and virion production. Visualizing these vRNA segments by FISH, especially all 8, is a powerful tool to study influenza selective assembly. The data presented supports the current model of IAV genome assembly where vRNP segments form bundles, the study also supports a newer model that the vRNP interaction clusters are flexible in nature as demonstrated with RNA-RNA interaction mapping in purified virions (PMID: 31332385). However, the study presented here has the advantage of knowing the precise three-dimensional location and composition of each vRNP bundle within a cell yet they do not utilize this information to further refine the vRNP networks at a cellular level. Given the recent data demonstrating that heterogeneity in viral infection at a single cell level (PMID 29451492 and PMID 30181264), the authors should consider whether the multiple bundling routes are a consequence of including cells that do not contain all 8 segments into their analysis.

Overall the visualization of all 8 segments within infected cells and in bound virions is unique and this technique serves as an additional strategy to confirm a model of influenza assembly where the vRNP form subcomplexes that are flexible as demonstrated by biochemical and recent machine learning approaches (PMID: 30689627). Overall, the clarity of the study could be improved with attention to the following issues.

Concern 1. The authors should provide experimental validation of the specificity for each probe set against other segments as well as any cRNA/vmRNA. Precise probe specificity is critical for accurate conclusions to be drawn from these data sets.

Response

We thank the reviewer for critically evaluating the experimental setup of our work. We agree that probe specificity is a critical issue. Therefore, probe specificity for vRNA as well as for mRNA probes was tested by several steps. First of all, Stellaris Probe Designer was used to generate a first set of FISH probes for both, vRNA and vmRNA, excluding human sequences by setting target organism "human". Afterwards, all FISH probes were again blasted against human sequences as well as against A/Panama vRNAs (taxid: 381513). A large quantity of FISH probes with highly similar matches against complementary human RNAs were excluded. Also, the vast majority of vRNA FISH probes of a particular segment did not show any relevant complementary sequence identity against other influenza genome segments, which means that the identity was less than 10 nucleotides. There were only a few probes among the initially over 300 vRNA FISH probes showing complete sequence identity to segments. Those were excluded for further use. However, even in case of binding of a single nucleotide to other segments, the fluorescence signal intensity would not be high enough to be detected via our fluorescence microscopy setup.

We also tested probe specificity of A/Panama probes by targeting A549 mock-infected cells, A549 cells infected with the influenza B/Lee virus strain, or VeroE6 cells infected with the Puumala hanta virus or (not shown). A fluorescence signal arising from unspecific binding

of FISH probes could not be detected in any of these control experiments. We have added this result to the main text (see Results paragraph '*Imaging of IAV vRNAs and vmRNAs*') and the respective figure to Supp. Infor. (see Fig. S5).

In addition, an antibody was used to measure cellular distribution of IAV NP (Fig. 1), which decorates vRNAs as well as cRNAs (for the latter see below). The NP pattern highly overlapped with the vRNA pattern and exhibited only very low similarity to the different vmRNA localization patterns again demonstrating specificity of the used vRNA FISH probes. Furthermore, we could not detect the formation of vmRNA MSCs of higher ranks (see Fig. S11) as found vRNA. These results are outlined in the Results paragraph '*Imaging of IAV vRNAs and vmRNAs*'.

With regards to distinction between vmRNA and cRNA, our vmRNA probes in principle should bind both RNAs. However, cRNAs were reported to be only present with at least 100fold lower copy numbers compared to vmRNAs and that cRNAs are predominantly present in the nucleus (Shapiro et al. J Virol, 1987. 61(3): p. 764-73). Taking into account the NP pattern, only vmRNA probe spots located in the nucleus and colocalizing with the NP signal could represent viral cRNAs, which was only the case for a minority of spots. This information and respective references are given in Supp. Info. (see '*Imaging of vmRNA*') indicated also by a link in the main text (Results paragraph '*Imaging of IAV vRNAs and vmRNAs*').

Concern

It is also important to note that the authors state that the colocalization algorithm was a custom script but have not publicly shared the code or imaging data sets.

Response

In the original version of our manuscript we outlined in the paragraph '*Colocalization analysis*' that we have used '..... a custom-written R script (R version 3.2.4) that will be made freely available upon request....' Following the request we have submitted via the file transfer site the respective software.

Concern 2. How do the networks defined by the authors in Fig 6 compare to the recently published RNA-RNA interactions for the Udorn or Wyoming/03/2003 H3N2 virus in PMID: 31332385. In particular, the lack of segments 7 and 8 in any of the initial complexes is intriguing and it may be due to the use of "all cells" in this analysis rather than cells that had an equal proportion of all 8 segments. Based on the data in Fig 3, segment 8 was under represented more often compared to the other segments and that may bias the networks constructed in Fig 6.

Response

We thank the reviewers for this advice concerning the comparison with recent data. Dadonaite et al. (PMID: 31332385) have used RNA-RNA cross-linking to show that there are numerous intra- and intersegmental RNA-RNA contacts. These lead to a complex, redundant and plastic network that allows most segments to interact with multiple other segments. Dadonaite et al. have shown for H1N1 (A/WSN/1933 and A/PR8/34) and H3N2 (A/Udorn/307/72) viruses that the network of interactions between closely related viruses is broadly similar, but there can be significant differences between more distant viruses. Since we also investigated a H3N2 strain (A/Panama), we can compare the results of Dadonaite et al. regarding H3N2 with ours. Dadonaite et al. found for S7 and S8 only a small number of interactions with other segments for H3N2 viruses (see Fig. 3 in Dadonaite et al.). In contrast, there were numerous interactions of segments S1 to S3 among each other and with other segments. These findings are consistent with our findings described above on the role of S1, S7 and S8 in MSC formation. Our results indicate that S1 does play a prominent role in the early phase of MSC formation while S7 and S8 not (see Fig.

6). The comparison of our results with those of Dadonaite et al. is now outlined in the last paragraph of '*Implications for IAV genome assembly*'.

We agree with the reviewer that a lower presence of a segment may influence the pathway of MSC formation. Here, the reviewer refers to S8 (Fig. 3A 'Total Segments'). Inspection of the data shows, that S8 is present in the same quantity as most of the other segments. We found only a small, but significant difference between S2 and S8 and S5 and S8 (see statistics, Tukey-test, legend to Fig. 3). However, there is no significant difference between S1 and S8. In contrast to S8, S1 is already present in the early phase of MSC formation, i.e. in forming MSCs of lower ranks (see Fig. 6). I.e., the somewhat lower abundance of S1 does not affect its presence in the early phase of MSC formation. For this reason, the slightly lower frequency of S8 cannot be responsible for the lack of this segment in the early phase of MSC formation.

Most relevant, we have also shown that an almost complete absence of S8 (Fig.3, Cell 29) does not prevent the formation of larger MSCs. In contrast, an almost complete absence of S1 (Fig.3, Cell 23) was associated with inhibition of the formation of MSC with higher ranks. These results as well as those of Daidonaite et al. (see above) suggest that a lack of S8 in the early phase of MSC formation is due to its low significance for this phase.

Concern 3. On page 8, the authors stated that, "we never obtained evidence for more than two copies of the same segment in an MSC; less than 5% of the MSCs contained two copies of the same segment". However, volume binning for vRNA spots in this study was 300 x 1000 nm, when they reference that a vRNP is 50-150 nm. If this statement was derived from the FISH experiments, it would be impossible that single vRNA of a specific segment could be differentiated given the limits of their methods and imaging resolution of standard confocal microscopy. This conclusion should be experimentally validated using approaches similar to PMID: 22547828 or else do not make such a claim.

Response

On behalf of the reviewer suggestion, we compared the colocalization radius to the more recent publication of Chou, et al. (Colocalization of different influenza viral RNA segments in the cytoplasm before viral budding as shown by single-molecule sensitivity FISH analysis. PLoS Pathog, 2013. 9(5): p. e1003358), because here the authors investigated also bundling of vRNAs in the cellular context. Here they have used a radius of 255 nm in x-y direction comparable to our radius of 300nm. Regarding the resolution of confocal fluorescence with roughly 200nm in x-y direction, these values were chosen in a reasonable way. Our 3D-stack acquisition was performed with 400 nm steps in z-direction due to the lower resolution in z-direction of confocal microscopy, which is the reason for the choice of 1000nm for colocalization analysis. The colocalization analysis revealed that less than 5% of all MSC spots showed presence of more than one segment copy per MSC. To evaluate these values and to probe whether multiple vRNAs are located within single fluorescence spots, STED microscopy in combination with "ordinary" confocal microscopy was performed. This experimental setup exhibited, that most of the single spots with confocal resolution remained single spots with STED resolution. We have modified/extended the following text (see '*Analysis of rank and of segment composition of MSCs*')

..... To relate the vRNA signals to each other we initially conducted a colocalization analysis binning two or more vRNA spots within a cylinder of radius 300 nm and height 1000 nm into one MSC. Although the size of isolated vRNP complexes varying in length between 50 to 150 nm³³ is smaller, these cylinder dimensions were chosen based on the limits of optical resolution and a previous study by Chou et al. ^{28,29} They have used a radius of 255 nm in x-y direction comparable to our cylinder radius and the resolution of confocal fluorescence with roughly 200 nm in x-y direction. Our 3D-stack acquisition in z-direction was performed with 400 nm steps due to the lower resolution in z-direction of confocal microscopy, which is the reason for choosing a height of 1000 nm for colocalization analysis.....

However, we agree with the view of the reviewer and follow the suggestion of the reviewer to remove this strong statement ("we never obtained evidence for more than two copies

of the same segment in an MSC") and replace it by the following sentence: "We found only in less than 5% indications for the presence of two, but not more copies of the same species. However, in view of the optical resolution achieved our approach does not allow to address accurately whether multiple copies of a segment are present in an MSC." (see paragraph '*Analysis of rank and of segment composition of MSCs*'). In addition, in the last sentences of this paragraph we extended our assumptions incl. citations for analysis of MSCs.

Concern 4.

The current model of IAV MSC assembly suggests that segments assemble as MSCs translocate from the nuclei to the plasma membrane. If so, lower ranked complexes would be observed near the nucleus with higher ranked complexes near the plasma membrane. The authors should analyze ranks of complexes within various distances from the nuclei and/or plasma membrane.

Response

We would like to assure the expert that we have intensively investigated and discussed this problem. But the answer to this question is problematic due to the morphology of the cell. The cells are very flat, i.e. the distance between the cell nucleus and plasma membrane is typically much greater in the x-y plane than in the z-direction. This raises the question of the reference site, i.e. distance to the plasma membrane in the z-plane or that of the x-y plane. However, since the direction of movement of the nascent vRNP bundle is not clear either, there are considerable uncertainties with regard to the evaluation. It would make sense to follow each individual growing vRNP bundle in its movement in the cytoplasm over time, which is not possible, however, due to the fixation of the cells that is necessary for this method. This recording of the trajectory is particularly important because it has not yet been conclusively clarified whether the movement from the cell nucleus to the plasma membrane is directed or can at least partially be chaotic or retrograde. In addition, we believe that parallel to the distance, a normalized volume must also be taken into account at the same time in order to be able to make a reliable statement. In fact, we have carried out an analysis of the occurrence of MSCs as a function of their size. From this analysis, we have been able to conclude that there is indeed a modest increase in the MSCs of higher ranks (especially rank 8) - in quantitative, but not in qualitative terms. This is also to be expected since these complexes become trapped during the formation of new viruses and are no longer (freely) mobile compared to the other complexes. Indeed, we have also found rank 8 complexes not far from the nucleus. Due to the limitations of such an analysis as described above and therefore the lack of reliable conclusions from our point of view, we have decided not to deal with this issue in the manuscript. We hope that the reviewer can understand our concerns.

Concern 5.

In Fig. 5C, it appears that the frequency of MSCs containing S1 is greater than S2 and so on (frequency of $S1 > S2 > S3 > \dots > S8$). In virions with MSCs ranked less than 8, is the probability of which vRNP segments make it into these particles linked to segment size? In this study, are larger segments found more often in lower ranked MSCs compared to smaller segments during infection? This could be an artifact of the number of probes capable of binding to a given segment and should be acknowledged in the text.

Response

To circumvent higher fluorescence signals arising from longer segments we used only similar probe numbers for each segment (~30 probes). However, due to the short length of segment 7 and 8, and due to our strict exclusion of probes after BLAST verification for probe specificity, only 22 or 21 probes could be generated for these segments. With regard to Fig. 3 (Total segments), we could demonstrate that this reduced probe numbers for S7 and S8 did not result in the observation of a strongly reduced quantity of S8 and especially

S8 in infected cells. Exactly the same probe sets were used for single virion studies suggesting that there should be no bias towards longer segment species. Furthermore, the higher frequency of the larger segments in MSCs is in agreement with the putative role of these segments in the early phase of MSC formation (see above, response to concern 2).

Concern 6.

In this study, the authors used a low-pathogenic avian IAV, A/Mallard, to study MSC assembly under non-permissive IAV infections. Given that infection with this virus is abortive in human A549 cells, the rationale for this experimental design is unclear. The replication kinetics of A/Mallard versus A/Panama are likely different and thus a 10 hpi cross-section between these infections may not be comparable. The complexity of this comparison makes these data difficult to interpret why the pattern of vRNP colocalization are different between these two virus infections.

Response

We agree with the reviewer that more details on virus infections are necessary to rationalize our results. We have recently shown (Sadewasser et al. (2017) Mol. Cell. Proteomics 16(5):728-742. doi: 10.1074/mcp.M116.065904) that

- A/Mallard and A/Panama enter A549 with similar efficiency
- both viruses express with similar efficiency viral proteins, although there were differences in the relative accumulation of M1 and NS
- only A/Panama efficiently replicate in A549, while replication of A/Mallard is one log step after 10 h p.i. and three log steps lower after 72 h p.i

In a follow up paper (Bogdanow et al. (2019) Nature Commun. 10(1):5518. doi: 10.1038/s41467-019-13520-8) we have shown

- no major difference between both strains with respect to A549 host protein and virus protein synthesis
- mRNA from both virus strains access the translational machinery with comparable efficiency, which argues against the idea that modulation of translation efficiency affects species specificity
- M1 and NA production of A/Mallard was lower than that of A/Panama, however, no differences in the global activity of RdRP activity between both strains
- non-permissive infection correlates with reduced M1 production and impaired nuclear export of NP and thus segment export;

In the revised version we have added the following paragraph (see last paragraph in *MSC assembly under non-permissive IAV infection*):

..... The reason for this observation needs future clarification. We have recently shown that replication of the avian A/Mallard/439/2004 virus in A549 cells is attenuated by three orders of magnitude at 72 h p.i. in comparison to A/Panama/2007/99^{42,43}. We were able to demonstrate that both viruses enter A549 with similar efficiency and that mRNA from both virus strains access the translational machinery with comparable efficiency. Interestingly the low productivity of A/Mallard infection in human cells correlated with reduced M1 production and impaired nuclear export of NP, which may interfere with the nuclear export of and interactions between vRNPs. In fact, while the nucleus of A/Panama infected cells had only a small number of segments (Figs. 1 and S1), the comparatively increased amount of some segments in the nucleus of A/Mallard infected cells (Fig. S10) may indicate a retarded export from the nucleus. However, also in this case further investigations are necessary for a validated conclusion. Furthermore, our recent results may also indicate that regular MSC assembly may not only depend on vRNP-vRNP interactions, but is co-regulated by host cell factors. We showed in a comparative quantitative proteomics a differential regulation of a number of host proteins in response to A/Panama vs. A/Mallard infections in A549 cells⁴². However, putative roles for these factors in the genesis and packaging of vRNPs remain to be investigated.....

Note

The following references mentioned by reviewer 1 are now cited:

PMID: 31332385

PMID: 29451492

PMID: 30689627

PMID: 22547828

Reviewer #2 (Remarks to the Author):

Haralampiev, Prisner, et al. used a multiplexed FISH assay to investigate all 8 influenza vRNA segments in parallel within A549 cells infected with human or avian influenza A viruses. These data support the prevailing model of influenza virus genome packaging in which each one of eight-influenza virus vRNPs are selectively bundled into multi-segment complexes and then subsequently incorporated into newly forming virions.

The authors developed significant improvements to an existing approach in order to answer an important and interesting question pertinent to the influenza field with relevance to other viruses with segmented genomes. The data have the potential to advance current understanding of influenza A virus genome assembly, an area of high interest in the field. However, a more rigorous analysis and systematic presentation of the data is greatly needed. The current version of the manuscript is unclear and poorly organized, to the extent that it is difficult to fully evaluate the results.

The authors should consider rectifying the following major and minor concerns:

Concern 1.

The introduction would benefit from a more comprehensive summary regarding several topics:

i) if you are going to present reassortment as an evolutionary benefit of genome segmentation, you should also mention fitness costs of reassortment

ii) packaging model—if you are going to mention both random vs selective packaging models, you should present evidence for the prevailing model (selective) but note the limitations of the previous studies,

iii) expand on the limitations of current understanding of influenza virus genome packaging mechanism and the signals that direct packaging.

Response

To i) We have included in the Introduction part a reference including a literature citation on the fitness costs of reassortment:

.... The segmented nature of the IAV genome can on one hand provide an evolutionary benefit as it enables the virus to evolve by reassortment of gene segments. On the contrary, reassortment may also bring together viral segments encoding proteins from parental strains, which work less well together thereby reducing viral fitness^{5,6}. (for a review see Lowen, A.C. 2017⁷).....

To ii)

Although we did not cover certainly limitations of all previous studies, we noted several of them without criticizing them. In the Introduction, the limitations of two papers were mentioned. In the first reference, the length of the segments was used as a rather indirect parameter for specificity of segments:

..... Using length as a parameter to distinguish between vRNP species, electron tomography of intact virions^{10,12} suggested that the viral genome is organized as an MSC with eight distinct segments, 7 of which are arranged around a central segment in a '7+1' pattern^{2,16}.

In the second reference, only four segments were visualized and detected by FISH:

..... Fluorescence *in situ* hybridization (FISH) studies^{28,29} resolving in parallel up to four out of the eight distinct vRNAs have provided strong evidence that these signals lead to formation of MSCs along the transit of the segments to the plasma membrane of infected cells.....

..... Fluorescence in situ hybridization (FISH) studies^{28,29} resolving in parallel up to four out of the eight distinct vRNAs have provided evidence that these signals lead to formation of MSCs along the transit of the segments to the plasma membrane of infected cells.....

Furthermore, we mentioned limitations of previous studies (see 'Analysis of rank and composition of MSCs')

..... Previous studies^{28,29} have reported an increase in the frequency of complexes with increasing rank in infected cells. At a first glance, this seems to be in contrast to the u-shaped distribution observed here. However, this apparent discrepancy to the earlier studies can easily be reconciled by the inherent limitation to resolve only a maximum of four distinct segments. Consequently, since no complexes of rank 5 or higher could be identified, MSCs observed as rank 1 to rank 4 include also complexes ranked higher than rank 4, leading the authors to suggest a continuous increase in the frequency of complexes with increasing rank....

In the light of text length restriction, we did not explore further the issue of limitations of previous studies.

To iii)

Following this recommendation, we have added the following part to the Introduction:

..... Support for selective packaging is given by identification of packaging signals in conserved 3' and 5' terminal non-coding (NCRs) and coding sequences (CRs) of the vRNAs, mediating interactions between vRNPs^{15,15,17-27}. Those CRs serve as segment bundling signals, whereas the NCRs operate as incorporation signals. Bundling sequences are proposed to interact with each other, whereby similar signals compete for integration and thus, ensure formation of segment bundles consisting of exactly eight different segments. Incorporation signals within the NCRs assure integration of the corresponding segment into progeny viruses, but all eight different NCR pairs need to be present for efficient virus growth....

..... Evidence for selective packaging includes the identification of packaging signals in conserved 3' and 5' terminal non-coding (NCRs) and coding regions (CRs) of the vRNAs, mediating interactions between vRNPs^{13,15,17-27}. Those CRs serve as segment bundling signals, whereas the NCRs operate as incorporation signals. Bundling sequences are proposed to interact with each other, whereby similar signals compete for integration and thus, ensure formation of bundles consisting of exactly eight different segments. Incorporation signals within the NCRs assure integration of the corresponding segment into progeny viruses, but all eight different NCR pairs need to be present for efficient virus growth....

Concern 2.

Quantification of observed differences were included throughout the manuscript. For example, avoid vague phrasing such as "mostly a rather balanced ratio", "approximately equal", "slightly lower than average" and "highly similar" and quantify these differences using statistics.

Response

We have removed respective phrases and have provided various statistical analysis including ANOVA (variance analysis), Bartlett-test (variability test), Tukey-test (multiple mean comparison), Chi-square test (frequency distribution) (see legends to Fig.2 and Fig.3), and normality test by D'Agostino-Pearson (see legend to Fig. S7).

Concern 3.

In sub-section 2 of the results section, expand on the rationale for determining vRNA spot dimensions and exclusion of nuclear spots. Also reference data to support the statement that <5% of MSCs showed 2 copies of the same segment.

Response

For determining vRNA spot dimension and the statement that <5% of MSCs showed 2 copies of the same segment, please see response to concern 3 of reviewer#1.

vRNA spots in the nucleus were excluded by the following reasons (see also revision of sub-section 1 of the Results section:

.... Nuclear vRNA spots were excluded for analysis for several reasons: i) low amounts of segments found in the nucleus if it all (Fig. 1), ii) vRNAs are produced inside the nucleus, meaning that also incomplete segments not capable of interacting with other segments may be detected by FISH, and iii) bundling is presumed to take place in the cytosol ²⁶....

.... Nuclear vRNA spots were excluded for analysis for several reasons: i) low amounts of segments found in the nucleus (Fig. 1), ii) incomplete vRNAs are produced inside the nucleus an detected by FISH may not be capable of interacting with other segments, and iii) bundling is presumed to take place in the cytosol²⁹....

Concern 4.

An important aspect of this study is the inclusion of human vs avian influenza A strains infecting a human cell line. The results would be more meaningful if presented in a way that cohesively and linearly presents these data (i.e., the figures should be presented chronologically in the text, e.g., figure 2C/D should be interpreted before analyzing Figure 3) with emphasis on comparing/contrasting human vs avian influenza A-infected cells.

Response

We agree with the reviewer that certainly an arrangement of the illustrations in accordance with the arrangement of the text would be beneficial for readability. However, we believe that the sequence we have chosen for the presentation of the results, i.e. first the results on vRNAs from A/Panama in A549 cells, then on vRNAs in intact A/Panama viruses and subsequently on vRNAs in A/Mallard in A549 cells, is reasonable. A parallel arrangement of the figures and the separate mapping of the results to A/Mallard would make a direct comparison with the results from A/Panama difficult. Furthermore, we are already at the limit of the permitted images and would otherwise exceed it significantly.

However, in following the reviewer's suggestion, we have made appropriate references in the text at the initial discussion of Figures 2, 3 and 4 to the later chapter on vRNAs in A/Mallard infected cells to support readability.

Concern 5.

Figure 3. Presentation of the data that shows the variance among the different cells (with statistical analysis) is needed. It is unclear whether row 1 represents an average over all cells (as mentioned in the text) or total number of MSCs or segments. Showing averages, with bars including error bars, would allow the authors to make meaningful comparisons between ranks rather than referring to the u-shaped distribution

Response

In addition to mean and variance of single segments (Fig. 2), we have now presented values summarizing 'All Cells' and 'Total Segments' (Figs. 3A and E) and data in Fig. S9 as mean \pm standard error/standard deviation.

We have also done an extensive statistical analysis (see response to Concern 2 of this reviewer).

Concern 6.

Fig. 4. Rather than showing segment counts for each MSC for each cell, it would be more impactful to consolidate these data to show major trends (or lack thereof) to more clearly articulate the conclusion that there is no dominant MSC composition within each rank.

Response

In fact, a limitation to single cell representation would not be meaningful whether there is a trend. Fig. 4A ('All Cells') shows the results from the entire cell population. This gives a clear trend, which is discussed in the Discussion section (see also Fig. 6).

Concern 7.

'Reliability of colocalization analysis' sub-section of the results should be included earlier on in the text. These are important controls that should be presented before the data, but do not necessarily need a separate section and can be integrated throughout the text.

Response

We can understand the reviewer's suggestion regarding the postponement of the section in question. However, at least part of the arguments presented in this section for the reliability of our analysis are based on findings and conclusions on the ranks and composition of MSCs etc. which have been presented in the upper part of the manuscript. In order to avoid a larger length of the manuscript, moving this section forward would require that reference be made to results and conclusions presented later in the manuscript. This would certainly not promote readability.

Since the other two reviewers did not make the point of a different arrangement of this section and this was not suggested in the editor's letter, we leave the original arrangement of the section in the manuscript.

Concern 8.

'Implications for IAV genome assembly' sub-section should be moved to the Discussion/Conclusions section of the manuscript.

Response

Done!

Concern 9.

Please include panel letters on all figures for clarity throughout the figure legends and text.

Response

Done!

Concern 10.

Figure legends should not contain results or interpretation of the data shown, except for a descriptive title if warranted.

Response

Following the reviewer's suggestion, we have revised and shortened the legends to Figures 1 and 3.

Concern 11.

Include more descriptive y axis titles for Figure 2A/C (e.g., Cells infected with A/Panama or Cells infected with A/Mallard).

Response

Done!

Concern 12.

Include axis labels for Fig. 5C. If represented on the same graph, scale of y-axis needs to be consistent. Purple color for NP vRNA is not distinguishable in image in Panel A.

Response

Done!

Reviewer #3 (Remarks to the Author):

The paper by Haralampiev et al. investigates the bundling of the 8 influenza A virus (IAV) genome segments by a multiplexed FISH assay. The authors could provide first insight into the cytoplasmic segment composition and possible genome complex intermediates. The paper describes for the first time a robust FISH-based assay to study genome bundling of all IAV segments and thus provides an excellent basis for future studies. However, besides developing a robust FISH assays, the authors did not really apply their method to address some open questions in the field of genome packaging.

Major points:**Concern 1.**

To convincingly demonstrate that the avian IAV used in this study is able to bundle correctly the authors should study the vRNP composition also in avian cells.

Response

The avian influenza A/mallard/Germany/439/2004 virus strain used in the study represents one of the many IAV circulating in their natural hosts. It can be grown to high titers in embryonated chicken eggs (about 1×10^7 PFU/ml) and there is no indication for an attenuated phenotype in avian cells. Hence, we assume that the segments of A/Mallard bundle regularly in avian cells, otherwise no reproduction of the virus can be expected. We do agree that it is of interest to study and confirm vRNP bundling in avian cells, but this analysis is out of the scope of the current project that focuses on processes in human cells, and will need to be investigated in a planned follow-up study

Concern 2.

Packaging sequences in the viral genome are required for correct packaging of the 8 different genome segments into a virus particle. It is believed that these sequences orchestrate genome bundling. Thus, it is not clear why the authors did not investigate a packaging mutant virus. This would provide valuable information.

Response

In order to characterize the significance of these packaging sequences for genome bundling, we follow two approaches: (i) investigation of reassortants, and (ii) introduction of mutations as suggested by the reviewer. We have generated a reassortante based on the strains used here. In the genome of A/Panama, the M-segment of A/Panama was replaced by the M-segment of A/Mallard, i.e. the reassortant used here contained the genome of A/Panama except for the M-segment which was supplied by A/Mallard. A549 cells were infected with this reassortant as described in the manuscript for A/Panama, and the analysis of the vRNA segments in the cells was performed 10 h p.i. The figure shows the frequency distribution of the MSC ranks. It is obvious that the formation of MSC of the higher ranks is significantly inhibited compared to A/Panama (see Fig. 3A in the manuscript) and is similar to the frequency distribution of the MSC ranks of A/Mallard (see Fig. 3E). This result shows that the formation of the complete genome in reassortants may be inhibited.

Abundances of MSCs in dependence on the complex rank for A/Panama with the M segment of A/Mallard (A/Pan-M). A549 cells were infected with A/Pan-M. With increasing complex rank, a decreasing population of MSCs was found for A/Pan-M similar to A/Mallard but very different to A/Panama (see Fig. 3).

A current focus of our investigations is the behavior of segment bundling in cells infected by reassortants, such as a recombinant expressing seven segments of Pan99 and the M segment of the avian Mal virus. Below we present first results on this topic.

Based on the relevance of the data for bundling, after completion of this rather extensive work, these investigations will be the subject of a separate publication. Furthermore, not least for reasons of manuscript length we won't include this topic into the present manuscript.

Concern 3.

There are published data available suggesting that vRNP bundling increases en route to the plasma membrane. This information is lacking and if technical possible should be included.

Response

See comment to concern 4 of reviewer 1.

Concern 4.

There are Panama virus-infected cells where 8 bundles are not efficiently formed, because some segments are poorly expressed. This suggests that balanced genome expression levels are important. To address this point experimentally the authors might consider to generate a virus that expresses less efficiently only one segment.

Response

We fully agree with the reviewer that the bundling behaviour in dependence of the absence or a very low expression of a vRNP allows detailed insights into the specific vRNP-vRNP interaction. We are currently carrying out extensive studies on this topic. To this end, we choose very low MOIs, which allows us to select cells that express a segment in very small amounts. However, these time consuming experiments are not yet completed and continuation is now interrupted due to SARS-CoV-2. The emerging results should not be presented in the Suppl. Infor. due to their relevance. Apart from the fact that we have not yet completed the study, the scope of this part would also be too extensive for the main text of the present manuscript, both from the point of view of the text and the additional illustrations.

However, we would like to point out that we have already observed and discussed exactly two such cases in the original manuscript (see Discussion). These were the low expression of S1 and S8. We could conclude that S1 plays an important role in the early phase of bundling, but S8 seems to be insignificant. We believe that we have thus provided examples that convincingly demonstrate the application of our approach to this question raised by the evaluator.

Minor points:

Concern 1.

Control experiments (page 15: Reliability....) is very difficult to understand.

Response

We apologize for any irritation. We hope that the revised text is now easy to understand.

Concern 2.

In the supplement all analyzed cells (Panama and Mallard) can be shown.

Response

We have now shown all analyzed cells of A/Panama and A/Mallard in an Excel data file (see 'Source Data File').

Concern 3.

Show pictures of genome distribution in Mallard-infected cells

Response

We now included in Supp. Information Fig. S10 showing the localization of vRNA and vmRNA in A/Mallard-infected A549 cells identified by MuSeq-FISH.

Concern 4.

Infection of human cells with the Mallard virus: are semi-infectious virus particles released from the cells?

Response

This is a valid question. The avian influenza A/Mallard/Germany/439/2004 virus used in the study represents one of the many IAV circulating in their natural hosts. It can be grown to high titers in embryonated chicken eggs (about 1×10^7 PFU/ml) and there is no indication for an attenuated phenotype in avian cells. The phenotypes of semi-infectious particles have to our knowledge mainly been studied with human IAV (nicely summarized by Diefenbacher et al, 2018), but we expected that this phenomenon – many virus particles fail to express all viral gene products – is also present in avian strains. Fig. 2 C and D show that single segments are clearly underrepresented in some of the cells infected with the avian virus, which could reflect infection by an SIP. We expect that this observation would also be reflected in the phenotypes produced by progeny viruses from these cells upon infection of "fresh" cells. Hence, our approach to determine MSC formation provides also a valuable model to decipher on of the several suggested mechanisms underlying SIP formation, but this can formally only be addressed in future analyses.

Concern 5.

Is the introduction of the human PB2 signature in the Mallard strain sufficient to restore genome bundling?

Response

Thank you for this question. In fact, we have tried hard during the revision of this work to establish a reverse genetic system for the Mal virus to be able to identify specific determinants of the apparent species-specific effect of MSC formation in human cells, but we were so far unable to recover respective mutant viruses. In the answer to concern 2, we show initial data on a 7+1 Panama reassortant virus, which suggest that the presence of the M segment from the avian virus disturbs regular vRNP bundling. Once, reverse genetics is feasible we will be able to assess contributions from other potential determinants to MSC formation such as the PB2 signature in future studies

Concern 6.

...mediating interactions between vRNPs (10,13-23)
Ref <https://doi.org/10.1073/pnas.0437772100> is missing.

Response

We thank the reviewer for pointing us to this reference which is now cited.

Concern 7.

...Fluorescence in situ hybridization (FISH) studies (24-26). Ref 24 is not accurate because that was FISH in particles.

Response

This has been corrected.

Concern 8.

Definition of packaging in the footnote: For the reviewer packaging is the incorporation of the genomes into viral particles.

Response

We thank the reviewer for the advice and have modified the footnote accordingly.

Concern 9.

...In fact, considering all possible combinations of vRNPs, the random packaging model predicts that 98% of the MSCs would contain two or more copies of a distinct segment species:

How do the authors come to 98%?

Response

The calculation is now given in Supplementary Information

Concern 10.

... To assess the vRNP composition of mature A/Panama virions by MuSeq-FISH, virions present in virus stocks were allowed to bind to the surface of human A549 cells on ice followed by immediate chemical fixation. Interestingly, the frequency distribution of MSCs in A/Panama virions was clearly different to that found in A/Panama-infected cells being shifted to MSCs of high rank (Fig. 5): How valid is the comparison? Which virus stock was used? One after infection with a high MOI or one after infection with a low MOI?

Response

Thank you for this question. We are clearly aware that the quality of the virus stocks used are critical in such experiments and we took any possible precaution to avoid generation

of defective interfering particles when growing Pan99 virus stocks in MDCK cultures. Hence, all experiments including detection of vRNPs by in situ hybridization inside infected cells, as well as in virions attached to the cell surface were done with plaque-purified Pan99 stock virus from infection with a multiplicity of 0.001.

Concern 11.

The authors analyzed only one time point post infection and one infection dose. Is the vRNP composition different at earlier time points and at lower MOIs?

Response

To address the request of the reviewer, we have done a similar experiment on infection of A549 cells by A/Panama at 6h p.i. (MOI 5) (see last paragraph in '*Analysis of rank and of segment composition of MSCs*'). The frequency distribution of the observed ranks of MSCs in the cytosol 6 h p.i. (Fig. S9) showed a left peaked form. It is obvious that the higher ranks at this time after infection are only present in small amounts. A comparison with the results obtained 10 h p.i. (Fig. 3) shows the dynamic character of the formation of the MSCs. Although further investigations are required, one reason for the reduced frequency of high rank MSCs may be due to the lower number of segments per cell (3.1×10^3 (6h p.i.) versus 5.6×10^3 (10h p.i.)).

Reviewers' Comments:

Reviewer #1:

Remarks to the Author:

In the revised manuscript entitled, "Selective, flexible packaging pathways of the segmented genome of Influenza A Virus", the authors utilized sequential FISH staining to visualize all 8 segments within single cells at a single timepoint (10hpi) and quantify the colocalization of segments. Visualizing all 8 vRNA segments is a powerful tool for studying IAV replication and assembly. The overall conclusions remain largely unchanged, 1) overall proportions of each segment are equal in a given cell, 2) the most abundant vRNP complexes are those with either 1 segment (rank 1) or 8 segments (rank 8), and 3) higher ranked complexes are found more often in intact virions.

Overall, the reviewers have revised their initial manuscript for clarity and have added the necessary caveats to the manuscript when describing the data. However, a few points should still be addressed.

1. The authors provided clarification on the infection of A/Mallard in A549 cells and revealed "impaired nuclear export of NP" (lines 264-266 and in the rebuttal letter), which would assume impaired nuclear export of vRNP as well. In addition, the authors report a significant defect in viral replication of A/mallard in A459 cells compare to A/Panama (ie a 3 log reduction in viral titer at 72 hpi). If vRNA segments are found in the cytoplasm, but NP nuclear export is defective that would also suggest that the NP - vRNA binding is altered in these cells and could then impact the RNA-RNA interactions.

Proper NP oligomerization, binding, and post-translational modifications are known to be important for vRNP packaging, so it unclear why that authors state that host proteins may be the restrictive feature (lines 271-273). While host proteins may be important for IAV vRNP assembly through a variety of mechanisms that could impact vRNP bundling, there is no data presented in this manuscript to support the assertion. Given the complexity of comparing data from A/mallard and A/Panama - the authors should remove the data as it is misleading to compare the two when the amount of cytoplasmic vRNP are not equal at this time point.

2. The authors provided the code for colocalization to the reviewers, but should make this freely available via github site to ensure that others can compare the colocalization algorithms. It is unclear, without extensive testing, whether the code uses ellipse centers for each segment spot to calculate the colocalization to nearby spots and whether the colocalization must fit all criteria - ie be within 300nm xy and 1000nm z or if a spot within 1000nm in z would still be counted in the MSC even if it was > than 300nm in xy.

3. In addition, the authors should acknowledge somewhere that not all 20-30 FISH probes will bind each segment. Chou et al (PMID: 22547828) very clearly demonstrate using photo-bleaching studies that only 7 FISH probes bound a single segment within a purified virion. This could be different within an infected cell as the accessibility of the FISH probe can change during assembly. This is the justification for why it is difficult to conclude that less than 2 copies of the same segment are located within a diffraction limited spot.

Reviewer #2:

Remarks to the Author:

Haralampiev et al. submitted a revised version of the manuscript titled "Selective flexible packaging pathways of the segmented genome of Influenza A virus" that includes improvements to data analysis and more comprehensive discussion. However, in its current state the narrative remains unclear.

- The application of statistics improved the analysis of the data (seen in figure legends), but the text describing the presentation and interpretation of results requires revision. The results section currently reads like a descriptive discussion. In some cases, the authors have included statistical analyses for parts of the figures and include explanation in the figure legends, but they do not use the statistics to make meaningful conclusions about the data. See examples in lines 132-141 and 185-188.

- Figures and figure panels should be labeled separately in the order of occurrence in the text of the manuscript. The authors need to restructure the text to address all panels of the figures in the order that they appear. It is disjointed and hard to follow in its current format.
- Lines 53-57 seem unnecessary to include since the paper does not address reassortment.
- Lines 62-65 appear to be contradictory and should be edited for clarity.
- Interpretation of data should not be included in the figure legends. Lines 785-787, 799-800, 825-827 should be moved to the appropriate results section. Statistical significance should be able to be determined from the figure itself (* or ns) with appropriate strata listed in the legends.
- Fig. 5c scale is incorrect (25 instead of 20). The colored bars are distracting, unless the color correlates to something else to improve understanding.

Reviewer #3:

Remarks to the Author:

Influenza A viruses co-package eight different genome segments into virions. Previous studies suggested that the eight genomic segments bundle inside infected cells prior to their co-packaging into new virus particles, however this process has remained to be visualized in detail. The current revised manuscript offers a novel FISH technique to analyze genome bundling inside infected cells. While the manuscript does not test whether previously suspected determinants such as vRNA-vRNA interactions among packaging signals are required for efficient genome bundling, it suggest three novel mechanisms:

1. balanced segment expression
2. balanced accumulation of vRNPs in the cytoplasm (nuclear export)
3. certain host factors

On the one hand, the data on A/Panama suggest that low expression levels of particular genome segments (segment 1 or segment 8) may abrogate the formation of high-rank MSCs at different stages of their assembly. The authors frankly admit that this hypothesis is only based on two analyzed cells.

Furthermore, the data on A/Mallard suggest that even in the presence of similar cytosolic levels of the eight segments, the formation of high-rank MSCs is inefficient. They further suggest that other mechanism(s) may influence genome bundling. This assumption is based on two older papers, where the authors identified that A/Mallard inefficiently exports the vRNPs from the nucleus to the cytoplasm and induces the expression of different host factors.

Taken together, all three mechanisms could affect genome bundling, yet the current data are too limited and vague to draw any conclusions. The manuscript requires additional data to support either of these three mechanisms.

The data provided by the authors in the refined manuscript suggest that the formation of high-rank MSCs is a dynamic process that changes with time. The authors provide a new experiment with A/Panama where they show that the formation of high-rank MSCs (Suppl. Fig 9) is very poor at 6 hpi, yet efficient at 10 hpi. In an older publication (Bogdanow et al., Nat. Com, 2019), the authors showed that at 6 hpi, A/Panama poorly exports its vRNPs from the nucleus, while they efficiently localize to the cytoplasm at 10 hpi. This suggests, that the accumulation of (certain levels of) the eight vRNPs in the cytoplasm triggers genome bundling. Importantly, this publication also revealed that the nuclear vRNP export of A/Mallard is substantially delayed compared to that of A/Panama. At 10 hpi, A/Mallard largely fails to export its genome from the nucleus, while at 16 hpi, the vRNPs partially accumulate in the cytoplasm. This may indicate that A/Mallard fails to bundle the genome as a consequence of the poor accumulation of vRNPs in the cytoplasm but is in principle able to bundle the genome when vRNPs are exported to the cytoplasm. To test this hypothesis, the authors should investigate MSC formation of A/Mallard at a later time point of infection. Ideally, the vRNPs should accumulate in the

cytoplasm to similar levels as observed with A/Panama at 10 hpi. If efficient vRNP accumulation triggers genome bundling, the eight MSC ranks would shift towards a U-shaped distribution. The authors are encouraged to perform this experiment as all experimental tools should be available. Furthermore, reviewer 1 has raised a similar concern (concern 6, reviewer 1).

Minor points:

- 1) line 268-269: "the comparatively increased amount": there is no quantification and only a picture is shown.
- 2) line 396-398: what is the rationale to compare RNA-RNA networks with MSC formation? Please specify.
- 3) line 413: what are transient segment shortages?

Reviewer #1

Concern 1

1. The authors provided clarification on the infection of A/Mallard in A549 cells and revealed "impaired nuclear export of NP" (lines 264-266 and in the rebuttal letter), which would assume impaired nuclear export of vRNP as well. In addition, the authors report a significant defect in viral replication of A/mallard in A459 cells compare to A/Panama (ie a 3 log reduction in viral titer at 72 hpi). If vRNA segments are found in the cytoplasm, but NP nuclear export is defective that would also suggest that the NP – vRNA binding is altered in these cells and could then impact the RNA-RNA interactions. Proper NP oligomerization, binding, and post-translational modifications are known to be important for vRNP packaging, so it unclear why that authors state that host proteins may be the restrictive feature (lines 271-273). While host proteins may be important for IAV vRNP assembly through a variety of mechanisms that could impact vRNP bundling, there is no data presented in this manuscript to support the assertion. Given the complexity of comparing data from A/mallard and A/Panama – the authors should remove the data as it is misleading to compare the two when the amount of cytoplasmic vRNP are not equal at this time point.

Response

Following the advice of the editor we have rephrased more carefully this paragraph and pointed out the limitations.

Concern 2

2. The authors provided the code for colocalization to the reviewers, but should make this freely available via github site to ensure that others can compare the colocalization algorithms. It is unclear, without extensive testing, whether the code uses ellipse centers for each segment spot to calculate the colocalization to nearby spots and whether the colocalization must fit all criteria - ie be within 300nm xy and 1000nm z or if a spot within 1000nm in z would still be counted in the MSC even if it was > than 300nm in xy.

Response

The code for colocalization was made available via GITHUB (<https://github.com/Budding-virus/Packbund>).

The spot must fil all mentioned criteria. This note has been now included in the Methods section (see Colocalization Analysis)

Concern 3

3. In addition, the authors should acknowledge somewhere that not all 20-30 FISH probes will bind each segment. Chou et al (PMID: 22547828) very clearly demonstrate using photo-bleaching studies that only 7 FISH probes bound a single segment within a purified virion. This could be different within an infected cell as the accessibility of the FISH probe can change during assembly. This is the justification for why it is difficult to conclude that less than 2 copies of the same segment are located within a diffraction limited spot.

Response

We have added this information to the paragraph 'Analysis of rank and of segment composition of MSCs' (lines 181-185):

... Furthermore, Chou et al.³⁵ have shown that not all specific FISH probes used bind to a segment of purified viruses. Thus, we cannot assume that all of the FISH probes we have used for a specific segment will bind. In particular, the accessibility of a segment could be additionally impacted in infected cells and the presence of more than one copy of the same segment might be missed.....

Reviewer #2

Concern 1

1. The application of statistics improved the analysis of the data (seen in figure legends), but the text describing the presentation and interpretation of results requires revision. The results section currently reads like a descriptive discussion. In some cases, the authors have included statistical analyses for parts of the figures and include explanation in the figure legends, but they do not use the statistics to make meaningful conclusions about the data. See examples in lines 132-141 and 185-188.

Former Lines 132-141:

Consistent with a previous report³², we observed cell-to-cell variability in the absolute number of individual segments (Supplementary Figure 7) and in the normalized number of individual segments as revealed by a statistical analysis (Fig. 2, legend, Bartlett-test). Nevertheless, at individual cell level a balanced ratio between the eight vRNA segments was observed (Fig. 2a). In a few cases, quantities of individual segments differed (Fig. 2a, Figs. 3c and d Cell 23 and 29). Overall, the population-averaged fractions of the various vRNAs were equal, with each appearing at about 12.5 % (Fig. 2b; statistics see legend). This agrees with qRT-PCR measurements at the cell population level (Supplementary Figure 8) showing that the expression levels of the different vRNA species were similar.

Former Lines 185-188

MSCs of high rank made up the majority of the total segments found (see 'Rank Distributions', Fig. 3, row 2, left, empty bars), and the number of solitary segments (rank 1) was low (see 'Free segments', Fig. 3, row 2, right). This observation strongly suggests that interactions between the segments direct towards the formation of high rank MSCs.

Response

Former Lines 132-141 have been modified accordingly (lines 147-153):

... Consistent with a previous report³², we observed cell-to-cell variability in the absolute number of individual segments and in the normalized number of individual segments as revealed by the frequency distribution of specific segments (Supplementary Figure 7) and statistical analysis (Fig. 2, legend, Bartlett-test), respectively. Nevertheless, for the majority of cells, at individual cell level a balanced ratio between the eight vRNA segments was observed (Fig. 2a) with no significant difference between the segments (ANOVA, F-test, for details see legend to Fig. 2).

Former Lines 185-188 have been also modified (lines 209-217):

.... MSCs of high rank made up the majority of the total segments found (Fig. 3 and b, empty bars), and the number of solitary segments (rank 1) was low (Fig. 3i and j) showing that the majority of each segment assembles into MSCs. This observation strongly suggests that interactions between the segments direct towards the formation of high rank MSCs. For a few cells (Cell 23 (Fig. 3c) and Cell 29 (Fig. 3d)), rank distributions were significantly different from those in Fig. 3a and b (Chi-square-test, $\alpha=0.05$; $P<0.0001$) which may provide clues for MSC assembly. Notably, the amounts of segment 8 (Cell 23 (Fig. 3g) and Cell 29 (Fig. 3h)) and segment 1 (Cell 23 (Fig. 3g)) were low in comparison to the remaining segments (see Discussion).

Concern 2

2. Figures and figure panels should be labeled separately in the order of occurrence in the text of the manuscript. The authors need to restructure the text to address all panels of the figures in the order that they appear. It is disjointed and hard to follow in its current format.

Response

We have improved labeling of figures/panels throughout the Results section of the main text and the Supplementary Information. In particular, data for A/Mallard presented in the former version in Fig. 2 and 3 have been moved now to a new Fig. 6.

Concern 3

3. Lines 53-57 seem unnecessary to include since the paper does not address reassortment.

Response

We are convinced that such investigations, as we present them in our manuscript, are important for the characterization and understanding of reassortment. Therefore, we do not want to delete this passage, especially since the other two reviewers have not criticized it. To support our view, we have added the following sentence (lines 61-63):
... Hence, knowledge of the mechanism(s) by which vRNPs assemble into a genome complex may support an improved understanding of biological phenotypes conferred by viral reassortment....

Concern 4

4. Lines 62-65 appear to be contradictory and should be edited for clarity.

Response

We do not understand the problem. In any case, to avoid a misunderstanding we have rephrased this part (lines 72-74):
...The different vRNPs interact specifically with each other and thus ensure the incorporation of exactly one copy of each segment into multi-segment complexes (MSC). Using length as a parameter to distinguish between vRNP species, electron tomography of intact virions^{10,12} suggested that the viral genome is organized as an MSC with eight different segments,...

Concern 5

5. Interpretation of data should not be included in the figure legends. Lines 785-787, 799-800, 825-827 should be moved to the appropriate results section. Statistical significance should be able to be determined from the figure itself (* or ns) with appropriate strata listed in the legends.

Response

We followed the recommendation of the reviewer. However, as statistical analysis was complex and detailed, essential parts we left to the figure legends. Moving the complex statistics to the main text would impair the readability.

Lines 785-787 have been removed and moved to the 2nd paragraph of the Results section (lines 128-129):

... vRNP segments (Fig. 1b, α -NP + vRNA). High degree of colocalization was observed for all vRNA segments (Fig. 1, vRNA overlay, white colouring). We tested.....

Lines 799-800

This information is now provided in the 3rd paragraph of the results (lines 150-153):
... Nevertheless, for the majority of cells, at individual cell level a balanced ratio between the eight vRNA segments was observed (Fig. 2a) with no significant difference between the segments (ANOVA, F-test, for details see legend to Fig. 2).

Lines 825-827

The lines 825-827 (legend of Fig. 3)

..... Statistics for 'Rank distributions' (Chi-square-test, $\alpha=0.05$): (a, b, e) The rank distributions are significant different from an uniform distribution ($P<0.0001$). (b, d) The rank distributions are significant different from each other ($P<0.0001$)....

have been moved to the 3rd and 4th paragraph of the subchapter 'Analysis of rank and of segment composition of MSCs'

Concern 6

6. Fig. 5c scale is incorrect (25 instead of 20). The colored bars are distracting, unless the color correlates to something else to improve understanding.

Response

The colour of the bars corresponds now to Fig. 4. The scale has been corrected.

Reviewer #3

The data provided by the authors in the refined manuscript suggest that the formation of high-rank MSCs is a dynamic process that changes with time. The authors provide a new experiment with A/Panama where they show that the formation of high-rank MSCs (Suppl. Fig 9) is very poor at 6 hpi, yet efficient at 10 hpi. In an older publication (Bogdanow et al., Nat. Com, 2019), the authors showed that at 6 hpi, A/Panama poorly exports its vRNPs from the nucleus, while they efficiently localize to the cytoplasm at 10 hpi. This suggests, that the accumulation of (certain levels of) the eight vRNPs in the cytoplasm triggers genome bundling. Importantly, this publication also revealed that the nuclear vRNP export of A/Mallard is substantially delayed compared to that of A/Panama. At 10 hpi, A/Mallard largely fails to export its genome from the nucleus, while at 16 hpi, the vRNPs partially accumulate in the cytoplasm. This may indicate that A/Mallard fails to bundle the genome as a consequence of the poor accumulation of vRNPs in the cytoplasm but is in principle able to bundle the genome when vRNPs are exported to the cytoplasm. To test this hypothesis, the authors should investigate MSC formation of A/Mallard at a later time point of infection. Ideally, the vRNPs should accumulate in the cytoplasm to similar levels as observed with A/Panama at 10 hpi. If efficient vRNP accumulation triggers genome bundling, the eight MSC ranks would shift towards a U-shaped distribution. The authors are encouraged to perform this experiment as all experimental tools should be available. Furthermore, reviewer 1 has raised a similar concern (concern 6, reviewer 1).

Response

Following the advice of the editor we have rephrased more carefully this paragraph and pointed out the limitations (lines 294-301). Such experiments are planned and will be performed after full recovery of lab activity. In any case, including this topic would dramatically exceed the content of this manuscript

Minor points:

Concern 1

1. line 268-269: "the comparatively increased amount": there is no quantification and only a picture is shown.

Response

We have omitted this sentence.

Concern 2

2. line 396-398: what is the rationale to compare RNA-RNA networks with MSC formation? Please specify.

Response:

We note that this comparison of RNA-RNA networks with MSC formation was suggested by reviewer #1 at the first review round of the manuscript.

We have added now a short specification in the last, concluding paragraph (lines 424-428 and 435-437):

... As RNA-RNA interactions are supposed to play a pivotal role in bundling of vRNP segments, it is of interest to address if our results on preferred vRNP compositions of intermediate MSCs are in agreement of intermolecular interactions between vRNAs which have been recently obtained by analysis of cross-linked vRNAs (Dadonaite et al.³⁷). This study has

..... Our findings on the role of S1 and S8 in MSC formation are consistent with their

observations, indicating a prominent role in the early phase of MSC formation of S1 but not of S8 (see Fig. 7).....

Concern 3

3. line 413: what are transient segment shortages?

Response

We have rephrased the sentence to clarify our point (lines 446-450):

.... However, maintaining some degree of redundancy within the genome packaging network confers robustness to the bundling process if several alternative segments were allowed to assemble at each given step.